# Escaping the Lock-in to Pesticide Use: Do Vietnamese Farmers Respond to Flower Strips as a Restoration Practice or Pest Management Action?

Finbarr G. Horgan [1,2,3,*], Quynh Vu [4,5,6], Enrique A. Mundaca [3], Shweta Dabholkar [2], Mark Davis [2], Josef Settele [6,7,8] and Eduardo Crisol-Martínez [1,3,9,*]

1. EcoLaVerna Integral Restoration Ecology, Bridestown, T56 P499 Kildinan, Ireland
2. Centre for Pesticide Suicide Prevention, University/BHF Centre for Cardiovascular Science, University of Edinburgh, Edinburgh EH16 4TJ, UK
3. Escuela de Agronomía, Facultad de Ciencias Agrarias y Forestales, Universidad Católica del Maule, Casilla 7-D, Curicó 3341695, Chile
4. Cuulong Delta Rice Research Institute, Tan Thanh, Thoi Lai District, Can Tho 94000, Vietnam
5. International Rice Research Institute, DAPO Box 7777, Metro Manila 1226, Philippines
6. Helmholtz Centre for Environmental Research—UFZ, Theodor-Lieser-Str. 4, 06120 Halle, Germany
7. German Centre for Integrative Biodiversity Research, Puschstrasse 4, 04103 Leipzig, Germany
8. Institute of Biological Sciences, University of the Philippines (UPLB), Los Baños 4031, Philippines
9. COEXPHAL (Association of Vegetable and Fruit Growers of Almeria), Carretera de Ronda 11, 04004 Almeria, Spain
* Correspondence: f.horgan@ecolaverna.org (F.G.H.); ecrisol@coexphal.es (E.C.-M.)

**Abstract:** Ecological engineering using linear flower strips is proposed as an alternative to insecticide-based rice pest management. However, its success depends on farmers' appreciations of related interventions as part of an ecosystem restoration process. We examined agronomic and pest management responses to flower strips among 305 farmers surveyed at 12 villages in the Mekong Delta Region (MDR) of Vietnam. Practices by conventional farmers at the same villages were used as a baseline. The ecological engineering farmers mainly integrated flower strips with pest management practices by reducing insecticide applications before 40 days after rice crop establishment (ca 38% of farmers; 9% more than on conventional farms). Flower strips were also associated with less frequent and irregular insecticide applications or with insecticide-free rice (i.e., possibly IPM: ca 19% of ecological engineering farmers). Otherwise, farmers (ca 43% of ecological engineering farmers) continued to apply insecticides prophylactically and, in some cases, applied more insecticides than their conventional neighbors. Flower strips were not associated with reductions in any other pesticides. Reported yields were not directly affected by flower strips or pesticide inputs. Our results suggest that ecological engineering was not widely regarded by participating farmers as an ecosystem restoration practice, but rather, as a pest management action. Further promotion of flower strips as a component of ecosystem restoration is required to break the lock-in to pesticide use at ecologically engineered rice farms in the MDR.

**Keywords:** agricultural policy; agrochemicals; agroecology; biological control; conservation agriculture; ecosystem resilience; integrated pest management; sustainable rice production

## 1. Introduction

Pesticide use among Asian farmers has increased dramatically since the early 2000s [1,2]. For example, pesticide use in some Asian countries increased by between 47% and 489% from 1990 to 2012, coinciding with enormous investments into Asia's chemical industries, particularly in India and China [2]. Furthermore, effective distribution and marketing networks have likely contributed to greater pesticide use among farmers, and vendors will often encourage Asian farmers to move from Integrated Pest Management (IPM) practices

toward prophylactic, calendar-based applications [3–5]. This shift to increasing and unnecessary insecticide use is attributed to farmers being locked-in to emerging crop production systems that promote a dependence on pesticides [3,4] and facilitate frequent interactions with proponents of chemical-intensive crop management [5]. Therefore, to avoid using pesticides and prevent consequent environmental and health problems, strategies are required to help farmers overcome pesticide lock-in.

An escape from pesticide lock-in will necessarily include mechanisms that alleviate the fear of pest-related yield losses (risk aversion) by educating farmers about the pest-regulating natural enemies that occur on their farms [6,7], as well as informing farmers about the harms caused by prophylactic (i.e., calendar or growth stage) applications [8]. Farmer field schools and campaigns around sustainable crop production have been somewhat successful in achieving this at different times and in different regions [8,9]. However, there is evidence that the achievements of such education campaigns can be quickly reversed in the face of lingering (mis)perceptions about pest-related yield losses [10] and intense marketing by agrochemical companies [4,5]. Such a regression toward intense pesticide use may be avoided where farmers are offered alternatives to pesticides, particularly where these alternatives are accompanied by education campaigns. In such cases, farmers' fears are abated through narratives around the natural regulation of pests, but also because the farmers become proactive in pest management (i.e., avoiding a 'do nothing' approach that is unlikely to be sustained where farmers' fears are constantly stoked). Alternatives to pesticide-intensive agriculture have included a range of novel preventative measures that reduce pest-related risks to crop production [11–21]; novel, environmentally friendly curative actions including botanical and microbial biopesticides together with modern application technologies that avoid economic losses by reducing pest densities below economic thresholds [22–26]; and improved decision support systems to better determine when curative actions are required [27,28].

In recent decades, considerable research attention has been directed toward ecological engineering as a preventative measure that supports rice pest management in Asia [11–18]. This agroecological approach to pest management aims to restore rice ecosystems by creating functional habitat that acts as a refuge for natural enemies [29–33]. Ecological engineering is mainly implemented using linear strips of flowering plants that provide nectar for the parasitoids and predators of rice pests, and habitat structure or alternative prey items for spiders [13,30]. One of the best examples of community-based ecological engineering for rice pest management has emerged in the Mekong Delta Region (MDR) of Vietnam [12,14,34–36]. With support from regional and provincial governments, thousands of rice farmers in the MDR have been encouraged to adopt linear flower strips as habitat for the natural enemies of rice pests [14,35]. This was accompanied by sponsored education campaigns that included demonstration plots and field days, as well as radio and TV soap operas [36,37]. To encourage adoption, government entities also distributed flower seeds to farmers for planting on bunds. Furthermore, farmers have been innovative in planting different flowering plants and many farmers have transitioned to growing flowering vegetable plants on their bunds to provide habitat for natural enemies while also gaining supplementary farm products [14]. Despite these advantages, a previous study has shown that MDR farmers who practiced ecological engineering had no greater appreciation for wildlife-related ecosystem services than their conventional farming neighbors [14]. Without an understanding of the ecosystem-related concepts underlying ecological engineering, a question arises as to whether farmers viewed flower strips as a systems approach to restoring pest-regulating ecosystem services (i.e., a restoration service from human society to the rice ecosystem [38]) or as a further action to be added to their current pest management practices.

In this study, we assess the agronomic and pest management responses by MDR rice farmers to flower strips. We propose that ecological engineering is best viewed as a holistic approach to ecosystem restoration (that includes pest regulation services) if farmers are to escape the lock-in to pesticide use resulting from intensification and the availability and

marketing of pesticides. For example, farmers, conscious of the need to conserve natural enemies might be expected to significantly reduce not only insecticide use but also the use of other toxic agrochemicals. Alternatively, we propose that farmers viewing ecological engineering as a simple pest management action—that is somehow equivalent to other interventions that reduce pest densities—are unlikely to escape the lock-in to pesticide use. Such farmers might substitute flower strips for some pest management intervention, such as a single insecticide application or the planting of resistant rice varieties. We, therefore, assessed information from surveys with ecological engineering and conventional farmers against defined indicators (see below) to assess how farmers have responded to the intervention. By comparing agronomic practices, and particularly the use of chemical pesticides, between conventional and ecological engineering farmers we determine how the farmers integrated flower strips with other pest management practices. We also identify the best predictors of farmers' pest management actions and their reported rice yields, based on farmer profiles, their use of fertilizers and other inputs, and their adoption of flower strips. Our approach separates IPM, as a decision system for curative actions, from agroecological interventions that aim to restore perturbed ecosystems and conserve natural enemies [36–38]. The approach will aid in clearly defining categories for alternatives based on their component roles in supporting farmers' transitions away from chemical pesticides. The paper also indicates how some of these different components interact. Therefore, based on our assessment of Vietnam's MDR, we make recommendations to gain further benefits from the adoption by Asian rice farmers of ecological engineering.

## 2. Methods

### 2.1. Integration Models for Flower Strips and Pest Management with Related Indicators

Ecological engineering in Vietnam, which was initiated in 2010, can be regarded as a movement that was coordinated among several stakeholder groups and focused on a specific issue related to rice pest management [14,36]. Based on contemporary publications, that 'issue' was the increasing frequency of pest outbreaks in chemical-intensive rice production systems across Asia [12,39,40]. In particular, a large body of research had linked outbreaks of phloem-feeding insects, such as the brown planthopper (*Nilaparvata lugens* (Stål)) and whitebacked planthopper (*Sogatella furcifera* Horvath), to a 'misuse' or 'overuse' of insecticides [40]. This was exacerbated wherever farmers applied large amounts of nitrogenous fertilizers [39,41]. The ecological engineering movement was based on the knowledge that insecticides deplete natural enemy numbers, thereby allowing herbivore densities to grow exponentially in the resulting enemy-free space [12]. Ecological engineering aimed specifically at restoring the regulating ecosystem services of natural enemies by providing habitat and food resources (e.g., nectar for the adult stages of specialist parasitoids or alternative prey for generalist predators) [30,32,33]. However, whereas depletion of natural regulation was regarded among the proximate causes of pest outbreaks, pesticides were the clear ultimate cause [40]. Therefore, for ecological engineering to be most effective, insecticide use would also have to be reduced or eliminated [12,17]. Indeed, it could be argued that a reduction in pesticide use was all that was required to restore regulating services to Vietnamese rice landscapes, but farmers were unlikely to do so without active involvement in the restoration process.

Based on this background, a '*basic conceptual*' model of integrating ecological engineering principles into existing pest management practices suggests that related interventions (mainly planting flower strips) would drastically reduce or eliminate insecticide applications if farmers adhere to the precepts of IPM—that is, using insecticides only as a last resource when pest densities approach economically damaging thresholds (model 1 in Figure 1). A similar model suggests that ecological engineering should increase system resilience against pest outbreaks in rice fields without pesticide use, including in '*organic and insecticide-free*' rice such that, compared to conventional farmers, those who establish flower strips on their rice bunds would be unlikely to consider insecticide use at any crop stage (model 2 in Figure 1).

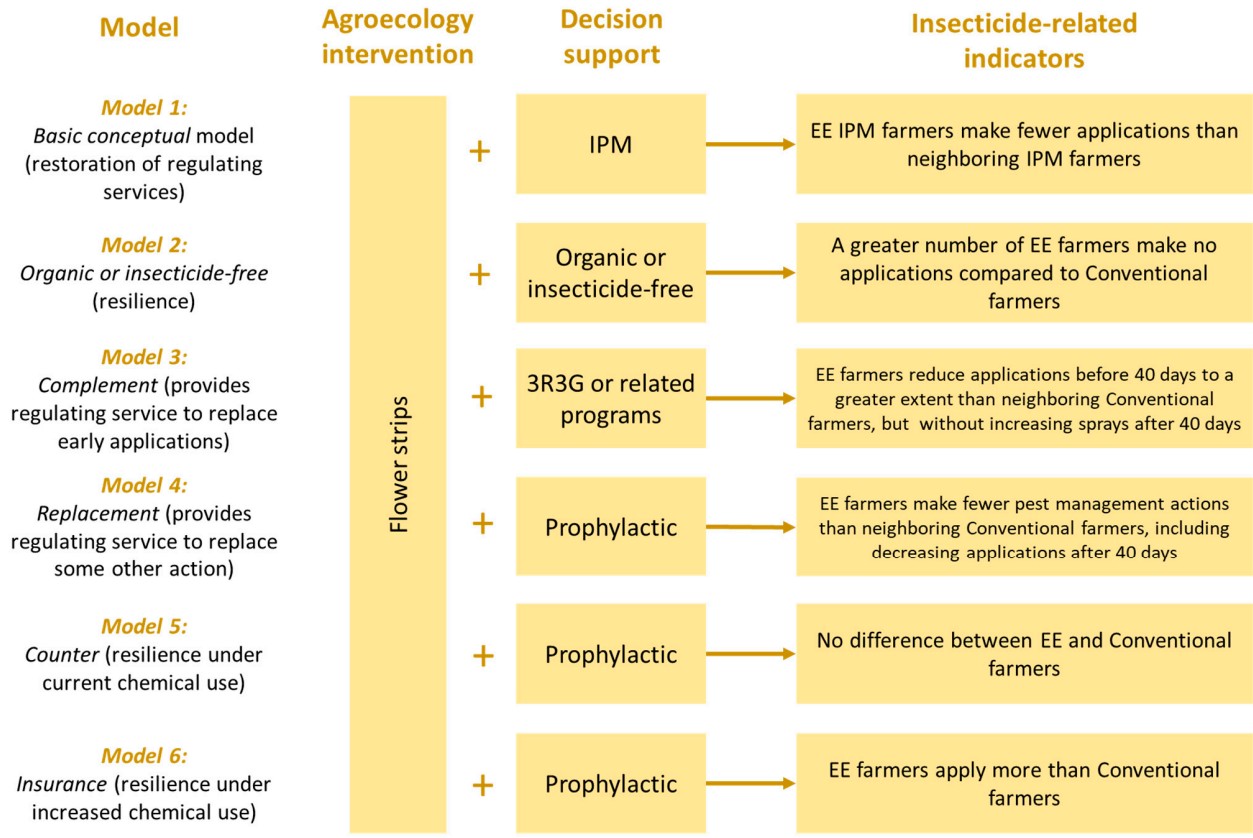

**Figure 1.** Models (scenarios) for the integration of ecological engineering (EE) using flower strips with pest management approaches (decision support for pesticide applications). A range of farmer responses (indicators) related to the number and timing of insecticide applications are used to identify which integration models the ecological engineering farmers predominantly follow.

Parallel rice management campaigns in the MDR, including 'Three Reductions, Three Gains' ('Ba Giam, Ba Tang' in Vietnamese—3R3G) and derived practices, have called for reductions in pesticide inputs and specifically recommend that insecticides not be used before 40 days after rice crop establishment [8,9,42]. This is based on the understanding that generalist natural enemies must build up numbers during early crop stages by consuming decomposer invertebrates and that specialist egg parasitoids are active during the early crop when rice is most vulnerable to immigrating planthoppers [41,43]. This recommendation was further promoted as part of the ecological engineering movement in the MDR—thereby complementing the 3R3G and related programs [37]. In this respect, ecological engineering might combat early prophylactic insecticide applications as a '*complement*' to existing rice management programs that encourage farmers to avoid applications before 40 days after rice crop establishment (model 3, Figure 1). Alternatively, the flower strips may be regarded as a '*replacement*' for other pest management actions, including crop rotations, host resistance, or insecticides (model 4, Figure 1). This latter, related, model is a special case of model 3, where farmers reduce insecticide use after 40 days while continuing prophylactic applications, or where they eliminate some other pest management practices such as the use of resistant varieties (Figure 1).

In at least one study, ecological engineering with insecticides was associated with increased densities of natural enemies compared to insecticide-based management without flower strips [12,17]. This suggests that some of the benefits of ecological engineering (i.e., increasing natural enemy abundance or efficiency) could be achieved even where farmers make prophylactic insecticide applications (model 5, Figure 1). In such a case, ecological engineering would only be adopted because it '*counters*' the negative effects of prophylactic applications (i.e., increasing resilience by preventing pest outbreaks in the

face of intense, prophylactic insecticide use), although it might also bring other benefits to the system including providing habitat for wildlife, especially pollinators [7,37,44,45]. One drawback of resilience—or the perception of resilience—provided by flower strips in intensive systems, is that it could lead to even greater levels of pesticide use if flower strips are regarded as an '*insurance*' (model 6, Figure 1) against pest resurgence, thereby allowing farmers to increase applications or apply insecticides indiscriminately.

To determine which model of ecological engineering farmers in the MDR were following, we assessed farmer responses during structured interviews against input indicators related to each model as presented in Figure 1. We used information from neighboring conventional farmers in the same villages or districts as a representative baseline for comparisons.

*2.2. Study Sites*

Surveys were conducted in the MDR in September 2015. The MDR produces over 40 million tons of rice annually, representing about 50% of Vietnam's total rice production and 70% of the nation's exported rice [46]. Rice production in the MDR is highly intensified. The region's tropical climate, year-round availability of water, and high availability of labor encourage rice triple-cropping [47]. The three growing seasons are referred to as đong xuân (winter–spring, harvested around February; henceforth 'season 1'), hè thu (harvested around June/July; henceforth 'season 2'), and mùa thu (harvested around October; henceforth 'season 3'). Some farmers rotate their rice with upland crops such as soya (*Glycine max* (L.) Merr.), maize (*Zea mays* L.), and horticultural crops [48]; and in flood-prone areas, many farmers also produce shrimp, crabs, or fish either in rotation with rice or incorporated into rice–fish systems [46,49]. Several villages in the MDR have adopted large-scale ecological engineering for pest management (known locally as 'công nghệ sinh thái') in rice production systems [14,35]. This consists of growing flowering ornamental or vegetable plants on the earthen bunds (levees) that transverse the rice habitat.

We interviewed farmers from nine districts across five provinces. Within each district, we identified villages with relatively high numbers of farmers growing flower or vegetable strips on their rice bunds. Farmers that had experience of ecological engineering, but were not currently practicing the method (i.e., 'method abandoned') are regarded here as conventional farmers because they did not practice ecological engineering during the seasons for which data were collected. Furthermore, according to Horgan et al. (2022) [14], these farmers have similar attitudes and use similar practices to other conventional farmers. Collaborators at relevant Provincial Plant Protection Departments coordinated with leaders at each sampled village to invite farmers to pre-established venues (usually a house associated with a farmers' association or a village hall) where the farmers were individually interviewed.

Within districts, we aimed to balance the numbers of farmers practicing conventional farming methods and ecological engineering; therefore, for some districts, we interviewed farmers from two or more villages to have representative samples within districts. The districts and villages were as follows: Châu Thành District (Tiền Giang Province), Tân Hội Đông (9 conventional, 18 ecological engineering) and Tân Lý Tây (11 conventional, 4 ecological engineering); Cai Lậy District (Tiền Giang Province), Tân Phú (4 conventional, 11 ecological engineering); Vũng Liêm District (Vinh Long Province), Hiếu Nhơn (3 conventional, 29 ecological engineering), Trung Hiếu (17 conventional), Hieu Thanh (6 conventional), Vũng Liêm (2 conventional) and Trung Thành Đông (3 conventional); Châu Thành District (An Giang Province), Bình Hòa (15 conventional, 7 ecological engineering) and Vĩnh Thành (6 conventional, 6 ecological engineering); Thoại Sơn District (An Giang Province), Vĩnh Khánh (10 conventional, 12 ecological engineering) and Vĩnh Trạch (3 conventional); Tân Hiệp District (Kiên Giang Province), Thạnh Đông A (19 conventional, 11 ecological engineering); Châu Thành District (Kiên Giang Province), Thạnh Lộc (19 conventional, 11 ecological engineering); Phước Long District (Bac Lieu Province), Vĩnh Phú Đông (21 conventional, 9 ecological engineering) and Phước Thành (5 conven-

tional, 5 ecological engineering); and Vĩnh Lợi District (Bạc Liêu Province), Vĩnh Hưng (16 conventional) and Châu Thới (4 conventional, 10 ecological engineering). Therefore, our sampled areas included both villages and village groups; the latter were considered where individual villages had a poor representation of either conventional or ecological engineering farmers. This allowed 12 representative comparisons. In all cases, village groups only included neighboring villages from a single district (Figure 2). To facilitate reading by non-Vietnamese-speakers, we substitute numbered references for each village name as indicated in the legend of Figure 2 and with the letter 'v' or 'g' to indicate a village of village group, respectively.

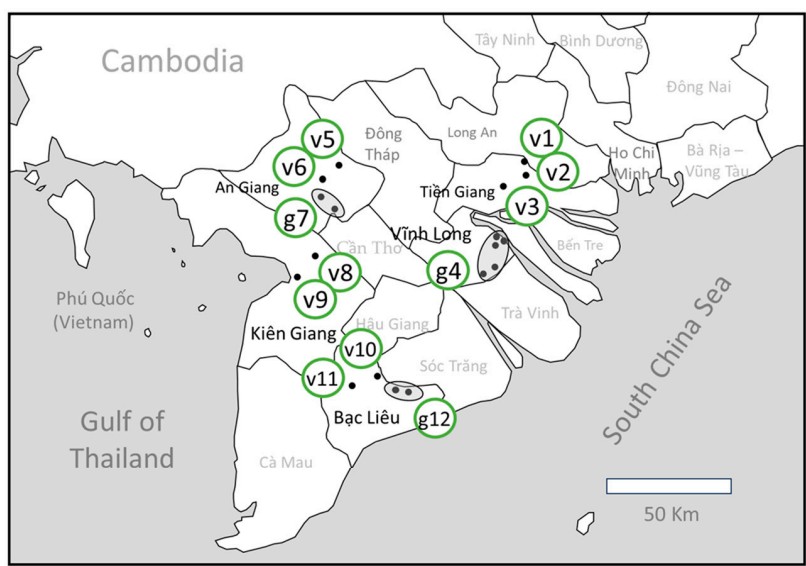
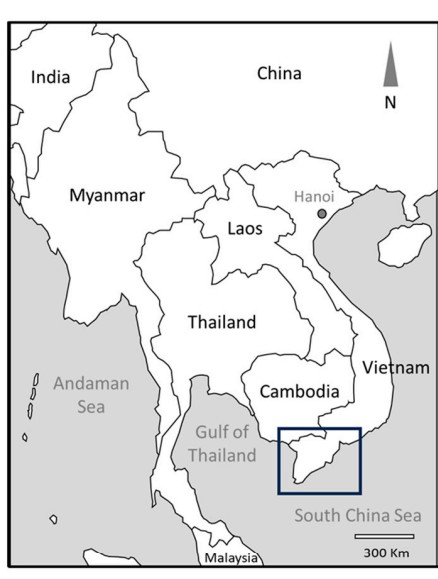

**Figure 2.** Map of mainland Southeast Asia (**right**) with the study region expanded (**left**) to indicate 12 surveyed sites in rice growing areas. Numbers indicate villages or village groups as follows: v1 = Tân Hội Đông, v2 = Tân Lý Tây, v3 = Tân Phú, g4 = Vũng Liêm group, v5 = Bình Hòa, v6 = Vĩnh Thành, g7 = Thoại Sơn group, v8 = Thạnh Đông A, v9 = Thạnh Lộc, v10 = Vĩnh Phú Đông, v11 = Phước Thành, and g12 = Vĩnh Lợi group. Figure adapted from Horgan et al. (2022) [14].

*2.3. Data Collection*

Each farmer was interviewed in Vietnamese by one of 20 trained interviewers using a structured questionnaire. No information was given to the village leaders (see above) or farmers regarding the questionnaire content prior to meeting the interviewers. The questionnaire was developed according to the knowledge, attitudes, and practices survey technique (KAP) [50], which allows a rapid appraisal of interventions in regions without pre-established researcher–community relations. The technique is also relatively resilient to varying interviewer experience levels. Further details about the survey, including some of the results as related to farmers' sources of knowledge concerning ecological engineering, their reasons for adopting the intervention, and their appreciations of wildlife and ecosystem services, are presented in a related paper [14]. This current paper uses the same database to present new information on village-level farming practices that has not been previously published.

The questionnaire was developed based on a focus group discussion (FGD) and pre-testing in the MDR [14]. We incorporated triangulation into the survey to improve information quality and to cross-check responses [51,52] and interviewers were also encouraged to record qualitative information during the interviews [53]. The final questionnaire consisted of four main parts: (1) farmer profiles, (2) the composition and management of flower or vegetable strips, (3) farmers' agronomic and pest management practices, and (4) farmer rice pest and disease concerns. The survey questions translated into English are presented in Table S1. The one-to-one interviews were conducted with 305 farmers and lasted approximately 12 min each.

Prior to conducting the interviews, the interviewers informed each farmer about the objectives of the interview, how the data would be used, and how the data would be stored (including that farmers' names would not be recorded and that reported results would not be linked to individual farmers). Farmers were also advised that they were not obliged to answer any questions.

*2.4. Data Analyses*

Farmers were divided into two groups (ecological engineering and conventional) across 12 villages/village groups for the analyses. Where responses to questions were nominal or ordinal, we conducted $2 \times 2$ contingency table analyses to test for associations with farmer or farm type and village; and three-dimensional table analyses to compare frequencies within categories (e.g., education levels, etc.). Tests of homogeneity and mutual and partial independence were conducted for all associations using $\chi^2$ analyses. Continuous dependent variables (e.g., reported inputs and yields) were initially analyzed using univariate general linear models (GLMs). The models examined the effects of farm type and village/village group and their interactions on farmer characteristics and practices. Where data were available for two or more seasons, the results were analyzed using repeated-measures GLMs. We used Tukey post hoc tests to assess homogenous farmer categories. Residuals were examined after all parametric analyses and were normal and homogenous. Univariate GLMs and contingency tables were analyzed using SPSS version 23.0 (IBM SPSS Statistics).

Three sets of appreciation variables were analyzed to (a) compare ecological engineering farmers' preferences for flowering plants and to understand the perceptions of farmers of both categories concerning (b) insecticide use (i.e., most commonly sprayed commercial insecticides) and (c) the most prevalent pests and diseases. The Permutational Analysis of Variance (PERMANOVA) [54] routine was used to test for differences between farm types and villages in each of the latter two sets of variables. Two factors were included in each analysis: 'farm type' was treated as a fixed factor with two levels (ecological engineering and conventional) and 'village' (fixed factor) with twelve levels, one for each village/village group surveyed. Whenever significant results were found, pair-wise tests were used to check for differences between levels. PERMANOVA analyses were based on Bray–Curtis similarity resemblance matrices of square root-transformed data; each analysis was permutated 9999 times. The similarity percentages routine (SIMPER) was applied to each of the three sets of variables to understand which variables contributed most to the generation of dissimilarities between pairs of groups [55]. Only the top contributing variables up to a cut-off of 75% dissimilarity were included in the SIMPER analyses. Canonical Analysis of Principal Coordinates (CAP) was used to visualize the differences in plant species composition on rice bunds [56]. PERMANOVA, SIMPER, and CAP routines were performed with PRIMER (v. 6.1.16) using the PERMANOVA + extension (v. 1.0.6).

The distance-based linear model (DistLM) routine [55] was used to identify which variables best predicted variations in yields and pesticide use. Six models were calculated, one for each dependent variable: yield, insecticide use, fungicide use, herbicide use, molluscicide use, and (total) pesticide use. A total of 17 predictor variables were included in each DistLM analysis, all related to farm management and farmer profiles (Table S2). Each DistLM was run using a stepwise routine based on the lowest AICc (Akaike's information

criterion corrected) selection criterion with 9999 permutations. DistLM analyses were performed using the PRIMER V6 statistical package with the PERMANOVA+ add-on (PRIMER-E Ltd., Plymouth, UK).

## 3. Results

### 3.1. Farmer Profiles

The farmers we interviewed were mainly male (< 6% were female). Conventional and ecological engineering farmers were similar in age ($F_{1,280}$ = 1.112, $p$ = 0.292), educational achievements ($\chi^2$ = 10.885, $p$ > 0.05), and farming experience ($F_{1,280}$ = 0.078, $p$ = 0.781) (Table S3). Rice farming was the main occupation of farmers in all villages ($\chi^2$ = 195.500, $p$ > 0.05) and for both types of farms ($\chi^2$ = 15.810, $p$ > 0.05) (Table S3). Among the villages where rice farmers also produced other crops (including horticultural and cereal crops), the frequencies of growing other crops were not different between villages ($\chi^2$ = 78.907, $p$ > 0.05) or between conventional and ecological engineering farmers ($\chi^2$ = 10.585, $p$ > 0.05) (Table S3).

Land ownership (i.e., the proportion of farmers owning their land) did not differ between villages or farm types. Farms were typically between 1 and 2 hectares. Farms tended to be smaller at v1, v2, and v3, and larger at v5, v6, and g7 ($F_{11,281}$ = 17.337, $p$ < 0.001). Ecologically engineered farms tended to be larger than conventional farms ($F_{1,281}$ = 5.994, $p$ = 0.015); however, there was a significant interaction between farm type and village because ecological engineering farmers at g4, v5, v6, and v7 had larger farms, but at v9, ecological engineering farms were smaller than the conventional farms (interaction: $F_{11,281}$ = 2.739, $p$ = 0.002) (Table S4). Further information on the profiles of farmers associated with each farm type, and the reasons that farmers adopted ecological engineering or not, has been presented by Horgan et al. (2022) [14].

### 3.2. Adoption of Ecological Engineering

Of the 305 farmers we interviewed, 133 were contemporaneously practicing ecological engineering by growing flowers or vegetables on their rice bunds. A further 36 farmers had attempted ecological engineering at some time in the past (mainly in the previous 1–5 years), but subsequently abandoned the practice (Table S4). Relatively large proportions of the farmers interviewed at v1, g7, and g12 had abandoned ecological engineering (Table S4). Farmers that still practiced ecological engineering had planted flowering plants on their bunds for 1 to 8 years, with farmers at v3 and v6 having more experience (>6 years) than farmers at v1 (1 year) ($F_{11,121}$ = 2.483, $p$ = 0.008: Table S4).

Details of bund management by farmers have been presented by Horgan et al. (2022) [14], including a list of the species grown on bunds, the methods of clearing bund weeds, and the end uses for the plant products. Here, we assessed the role of village/village group on the composition of flower strips. There was a significant village effect (F = 5.364, $p$ < 0.001) (Figure 3) on the choice of plant species by ecological engineering farmers, with villages/village groups v1, g4, v10, and g12 different from every other village (Table S5). Based on the SIMPER test results, the predominant species at the latter four villages were Cosmos (*Cosmos* spp.), Butterdaisy (*Melampodium* spp.), or Lantana (*Lantana camara* L.). These three species contributed most to the differences in species composition across all villages (Figure 3, Table S5).

### 3.3. Agronomic Practices

Rice was triple-cropped by most farmers, except at v9, v10, and v11, where many farmers planted one or two rice crops per year (Figure S1) sometimes rotated with field or vegetable crops (Table 1). Fallows between crops were generally less than 30 days, and often less than 15 days (Table 1). Rice was mainly direct seeded at <5 days after germination with 16% of farmers using machine sowing (mainly farmers at v5, v6, and g7 and the ecological engineering farmers at g4). Only 5.5% of farmers, mainly at v1 (33%), transplanted their rice as <20-day-old seedlings during at least one season (Table 1).

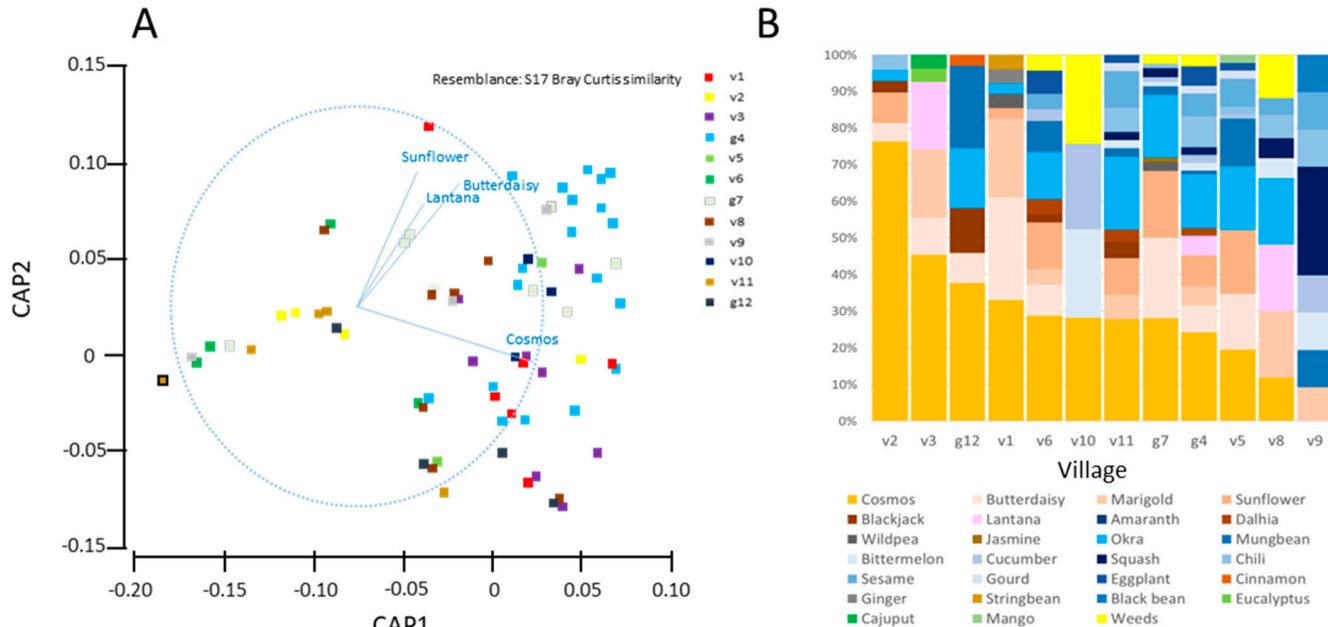

**Figure 3.** (**A**) Canonical Analysis of Principal Coordinates (CAP) showing differences between villages in plant species composition on rice bunds. Overlaid vectors (in blue) represent the correlations (Pearson's correlation coefficient > 0.6) between plant species and the CAP axes, where vector length and direction reflect the increasing values of correlation and parameter values, respectively. (**B**) The proportional representation (i.e., number of farmers planting the species/number of farmers × number of species) of different plant species on bunds is indicated for each village. Flowering ornamental plants are indicated by various shades of orange, flowering vegetable plants by various shades of blue, trees by green shading, and spontaneous weeds in yellow. Scientific names are presented in Table S6.

**Table 1.** Agronomic practices adopted by ecological engineering and conventional rice farmers at 12 villages/village groups in the MDR.

| Village [1] | Farm Type [2] | Season [3] | Rotating Crops (%) | Length of Fallow (Days) [4] | Age of Seedlings (Days) [4] | Nitrogen (Kg ha$^{-1}$) [4] | Potassium (Kg ha$^{-1}$) [4] | Phosphorus (Kg ha$^{-1}$) [4] | Perceived Yields (Tons ha$^{-1}$) [4] |
|---|---|---|---|---|---|---|---|---|---|
| v1 | C | 1 | 11.11 | 15.56 (2.32) [abc] | 5.56 (1.27) [b] | 66.97 (12.66) [abc] | 33.90 (6.18) [bc] | 49.84 (12.26) [ab] | 5.55 (6.20) [bc] |
| | | 2 | | 21.00 (2.51) | 5.56 (1.27) | 53.51 (15.57) | 33.56 (4.51) | 28.73 (6.94) | 8.22 (3.29) |
| | | 3 | | 17.11 (2.08) | 5.56 (1.27) | 57.68 (16.80) | 32.49 (4.95) | 26.34 (7.92) | 5.45 (7.23) |
| | EE | 1 | 11.11 | 17.28 (1.17) | 5.89 (0.78) | 99.60 (10.17) | 37.31 (5.49) | 72.25 (14.45) | 6.07 (9.16) |
| | | 2 | | 25.33 (3.37) | 5.89 (0.78) | 92.93 (8.62) | 40.27 (5.34) | 73.21 (13.80) | 7.94 (2.22) |
| | | 3 | | 21.61 (2.63) | 5.89 (0.78) | 89.83 (7.88) | 40.08 (5.61) | 60.13 (10.27) | 6.06 (9.26) |
| v2 | C | 1 | 20.00 | 13.00 (1.33) [a] | 3.25 (0.25) [a] | 79.51 (9.22) [abc] | 27.90 (4.91) [b] | 35.21 (8.64) [a] | 4.88 (9.42) [a] |
| | | 2 | | 16.11 (2.00) | 3.25 (0.25) | 86.48 (12.37) | 38.17 (12.46) | 51.29 (19.88) | 5.87 (6.38) |
| | | 3 | | 14.44 (1.00) | 3.25 (0.25) | 82.14 (9.86) | 27.90 (4.91) | 35.98 (8.59) | 4.71 (2.33) |
| | EE | 1 | 50.00 | 17.50 (2.50) | 3.00 (0.00) | 103.81 (8.89) | 26.75 (3.75) | 68.75 (20.25) | 5.59 (0.43) |

Table 1. *Cont.*

| Village [1] | Farm Type [2] | Season [3] | Rotating Crops (%) | Length of Fallow (Days) [4] | Age of Seedlings (Days) [4] | Nitrogen (Kg ha$^{-1}$) [4] | Potassium (Kg ha$^{-1}$) [4] | Phosphorus (Kg ha$^{-1}$) [4] | Perceived Yields (Tons ha$^{-1}$) [4] |
|---|---|---|---|---|---|---|---|---|---|
| | | 2 | | 16.25 (1.25) | 14.00 (11.00) | 103.81 (8.89) | 26.75 (3.75) | 68.75 (20.25) | 6.24 (0.41) |
| | | 3 | | 17.50 (2.50) | 3.00 (0.00) | 103.81 (8.89) | 26.75 (3.75) | 68.75 (20.25) | 6.02 (0.86) |
| v3 | C | 1 | 25.00 | 21.25 (5.15) [c] | 3.25 (0.25) [a] | 45.33 (3.40) [ab] | 26.39 (11.47) [bc] | 33.07 (1.72) [a] | 6.77 (5.67) [bc] |
| | | 2 | | 60.00 (0.00) | 3.25 (0.25) | 44.01 (4.31) | 24.18 (11.95) | 30.86 (2.31) | 8.60 (4.00) |
| | | 3 | | 11.25 (1.25) | 3.25 (0.25) | 43.94 (4.36) | 24.07 (11.98) | 30.75 (2.38) | 6.67 (3.22) |
| | EE | 1 | 27.27 | 18.64 (2.25) | 3.00 (0.00) | 88.06 (6.46) | 40.77 (5.66) | 51.03 (4.34) | 5.29 (8.14) |
| | | 2 | | 40.91 (4.56) | 3.00 (0.00) | 77.03 (8.65) | 41.15 (6.71) | 45.58 (5.52) | 7.70 (7.26) |
| | | 3 | | 12.91 (0.91) | 3.00 (0.00) | 89.14 (5.87) | 41.87 (5.89) | 50.88 (4.36) | 6.18 (9.24) |
| g4 | C | 1 | 3.23 | 20.65 (0.95) [bc] | 2.84 (0.07) [a] | 86.59 (3.49) [abc] | 35.59 (3.01) [bc] | 72.57 (2.92) [ab] | 5.47 (6.10) [bc] |
| | | 2 | | 29.84 (1.68) | 2.84 (0.07) | 85.51 (3.52) | 30.51 (1.97) | 72.92 (2.90) | 6.70 (5.13) |
| | | 3 | | 20.00 (0.40) | 2.84 (0.07) | 85.79 (4.88) | 34.16 (4.83) | 74.08 (5.02) | 5.46 (5.09) |
| | EE | 1 | 0.00 | 16.72 (0.45) | 2.79 (0.43) | 95.61 (3.14) | 43.44 (3.25) | 60.85 (5.69) | 6.30 (5.11) |
| | | 2 | | 24.31 (1.31) | 2.79 (0.43) | 90.36 (3.69) | 49.55 (4.74) | 51.90 (2.96) | 7.78 (0.21) |
| | | 3 | | 18.45 (0.56) | 2.79 (0.43) | 91.05 (3.28) | 42.47 (3.40) | 52.00 (4.03) | 6.47 (1.14) |
| v5 | C | 1 | 0.00 | 15.53 (1.40) [ab] | 3.93 (0.64) [ab] | 129.69 (12.41) [d] | 54.56 (6.76) [bc] | 63.99 (6.46) [ab] | 5.88 (2.14) [b] |
| | | 2 | | 18.33 (1.67) | 3.93 (0.64) | 135.95 (10.64) | 51.08 (6.83) | 63.62 (6.31) | 6.93 (3.19) |
| | | 3 | | 18.33 (1.05) | 3.93 (0.64) | 135.67 (10.78) | 51.96 (7.00) | 62.70 (6.60) | 5.95 (3.19) |
| | EE | 1 | 14.29 | 20.00 (0.00) | 6.00 (1.41) | 130.18 (8.96) | 47.93 (6.55) | 54.05 (4.49) | 5.47 (3.37) |
| | | 2 | | 29.29 (5.39) | 5.57 (1.27) | 134.20 (7.60) | 55.30 (6.54) | 58.03 (4.53) | 6.77 (3.21) |
| | | 3 | | 19.29 (0.71) | 5.57 (1.27) | 129.37 (7.61) | 51.02 (5.08) | 57.98 (4.53) | 5.78 (2.14) |
| v6 | C | 1 | 0.00 | 14.00 (1.53) [ab] | 4.00 (1.21) [ab] | 94.98 (19.96) [bc] | 54.87 (14.87) [bc] | 51.86 (6.95) [ab] | 5.63 (4.12) [bc] |
| | | 2 | | 16.67 (1.67) | 4.00 (1.21) | 104.64 (21.77) | 54.87 (14.87) | 55.06 (8.11) | 7.12 (0.37) |
| | | 3 | | 16.67 (1.05) | 4.00 (1.21) | 101.23 (21.22) | 58.03 (14.25) | 67.83 (18.42) | 5.42 (0.22) |
| | EE | 1 | 0.00 | 15.00 (1.83) | 5.17 (1.54) | 84.88 (11.44) | 41.95 (7.59) | 53.77 (8.66) | 5.55 (4.34) |
| | | 2 | | 20.83 (2.39) | 5.17 (1.54) | 104.44 (12.10) | 43.38 (7.29) | 63.30 (8.25) | 7.74 (0.35) |
| | | 3 | | 17.50 (1.12) | 5.17 (1.54) | 103.08 (12.56) | 43.06 (7.42) | 59.99 (7.50) | 6.08 (0.23) |
| g7 | C | 1 | 0.00 | 22.77 (1.45) [ab] | 3.00 (0.00) [ab] | 107.57 (8.25) [cd] | 50.16 (5.15) [c] | 78.27 (9.86) [b] | 6.13 (0.26) [c] |
| | | 2 | | 20.08 (0.98) | 3.69 (0.69) | 107.59 (8.87) | 50.16 (5.15) | 82.88 (12.89) | 7.93 (0.24) |
| | | 3 | | 22.00 (1.33) | 3.69 (0.69) | 108.57 (8.48) | 52.60 (6.52) | 78.98 (10.11) | 5.93 (0.22) |
| | EE | 1 | 0.00 | 15.75 (1.21) | 4.17 (1.06) | 121.07 (6.57) | 53.09 (7.28) | 91.47 (8.75) | 6.92 (2.22) |
| | | 2 | | 18.25 (1.57) | 4.17 (1.06) | 121.07 (6.57) | 66.14 (19.35) | 91.47 (8.75) | 8.36 (2.25) |
| | | 3 | | 16.00 (1.66) | 4.17 (1.06) | 121.07 (6.57) | 50.59 (7.79) | 91.47 (8.75) | 6.28 (2.30) |

Table 1. *Cont.*

| Village [1] | Farm Type [2] | Season [3] | Rotating Crops (%) | Length of Fallow (Days) [4] | Age of Seedlings (Days) [4] | Nitrogen (Kg ha$^{-1}$) [4] | Potassium (Kg ha$^{-1}$) [4] | Phosphorus (Kg ha$^{-1}$) [4] | Perceived Yields (Tons ha$^{-1}$) [4] |
|---|---|---|---|---|---|---|---|---|---|
| v8 | C | 1 | 0.00 | 21.00 (0.87) [bc] | 3.00 (0.00) [a] | 63.44 (2.04) [a] | 31.72 (1.02) [bc] | 63.44 (2.04) [ab] | 5.81 (3.27) [b] |
|  |  | 2 |  | 21.05 (0.72) | 3.00 (0.00) | 69.87 (1.89) | 34.93 (0.95) | 69.87 (1.89) | 7.05 (9.24) |
|  |  | 3 |  | 21.58 (0.77) | 3.00 (0.00) | 62.48 (1.22) | 31.24 (0.61) | 62.48 (1.22) | 5.80 (2.16) |
|  | EE | 1 | 0.00 | 19.50 (0.90) | 3.00 (0.00) | 65.60 (2.32) | 32.80 (1.16) | 65.60 (2.32) | 5.53 (5.16) |
|  |  | 2 |  | 26.36 (3.57) | 3.00 (0.00) | 72.73 (2.28) | 36.36 (1.14) | 72.73 (2.28) | 6.88 (2.27) |
|  |  | 3 |  | 19.55 (0.81) | 3.00 (0.00) | 65.45 (1.81) | 32.73 (0.91) | 65.45 (1.81) | 5.73 (5.11) |
| v97 | C | 1 | 0.00 | - | - | - | - | - | - |
|  |  | 2 |  | 26.94 (1.67) [bc] | 3.00 (0.00) [a] | 63.22 (3.52) [a] | 31.61 (1.76) [bc] | 63.22 (3.52) [ab] | 6.79 (3.20) [b] |
|  |  | 3 |  | 19.72 (0.28) | 3.00 (0.00) | 59.52 (2.52) | 29.76 (1.26) | 59.52 (2.52) | 5.99 (9.21) |
|  | EE | 1 | 0.00 | - | - | - | - | - | - |
|  |  | 2 |  | 27.27 (3.53) | 3.00 (0.00) | 66.91 (5.53) | 33.45 (2.76) | 66.91 (5.53) | 6.32 (6.16) |
|  |  | 3 |  | 19.09 (0.61) | 3.00 (0.00) | 68.36 (4.75) | 34.18 (2.37) | 68.36 (4.75) | 5.80 (8.27) |
| v10 | C | 1 | 0.00 | 28.00 (4.36) [d] | 3.00 (0.00) [a] | 89.76 (5.50) [ab] | 44.50 (6.14) [bc] | 75.93 (10.95) [ab] | 5.83 (9.17) [bc] |
|  |  | 2 |  | 76.92 (11.76) | 3.00 (0.00) | 80.43 (7.25) | 36.74 (5.09) | 81.93 (10.90) | 7.76 (0.17) |
|  |  | 3 |  | 55.00 (6.35) | 3.00 (0.00) | 42.17 (8.06) | 26.24 (4.28) | 45.21 (9.03) | 5.78 (2.14) |
|  | EE | 1 | 11.11 | - | - | - | - | - | - |
|  |  | 2 |  | 87.86 (17.14) | 3.00 (0.00) | 70.87 (11.26) | 37.85 (11.08) | 55.84 (17.21) | 7.29 (0.15) |
|  |  | 3 |  | 79.29 (9.09) | 3.00 (0.00) | 66.04 (11.17) | 29.66 (10.79) | 53.11 (15.43) | 5.32 (6.23) |
| v117 | C | 1 | 100.00 | 30.00 (0.00) [e] | 3.00 (0.00) [a] | 104.83 (17.58) [bc] | 11.33 (2.64) [a] | 43.53 (12.60) [ab] | 6.33 (0.27) [bc] |
|  |  | 2 |  | - | - | - | - | - | - |
|  |  | 3 |  | - | - | - | - | - | - |
|  | EE | 1 | 100.00 | 30.00 (0.00) | 3.00 (0.00) | 82.76 (9.92) | 13.50 (1.00) | 73.30 (5.62) | 6.50 (2.31) |
|  |  | 2 |  | - | - | - | - | - | - |
|  |  | 3 |  | - | - | - | - | - | - |
| g12 | C | 1 | 0.00 | 14.75 (0.57) [ab] | 3.05 (0.14) [a] | 93.76 (7.69) [abc] | 45.35 (5.42) [bc] | 60.96 (5.62) [ab] | 5.97 (3.11) [bc] |
|  |  | 2 |  | 16.10 (1.19) | 3.10 (0.16) | 112.63 (9.07) | 49.08 (5.71) | 67.89 (5.87) | 6.91 (1.13) |
|  |  | 3 |  | 23.75 (1.53) | 3.05 (0.14) | 98.09 (8.03) | 44.36 (5.17) | 61.54 (5.70) | 6.54 (8.13) |
|  | EE | 1 | 0.00 | 16.00 (1.00) | 3.00 (0.00) | 51.25 (11.78) | 20.18 (4.66) | 48.72 (15.18) | 5.64 (1.17) |
|  |  | 2 |  | 19.50 (2.03) | 3.00 (0.00) | 70.32 (19.21) | 22.69 (3.97) | 70.51 (24.29) | 7.00 (0.20) |
|  |  | 3 |  | 20.50 (3.20) | 3.00 (0.00) | 49.77 (11.56) | 20.16 (4.75) | 44.79 (11.27) | 6.07 (9.27) |
| Season [5,7] |  |  |  | 57.726 *** | 0.371 ns | 5.086 ** | 2.816 ns | 3.999 * | 319.341 *** |
| Season × Village [5,7] |  |  |  | 13.509 *** | 0.807 ns | 4.028 *** | 0.523 ns | 2.089 ** | 6.174 *** |
| Season × Farm type [5,7] |  |  |  | 0.176 ns | 0.490 ns | 0.034 ns | 0.677 ns | 0.150 ns | 2.029 ns |
| Season × Village × Farm type [5,7] |  |  |  | 2.663 *** | 0.891 ns | 0.282 ns | 0.698 ns | 0.754 ns | 1.647* |
| Village [6] |  |  |  | 9.558 *** | 5.332 *** | 10.936 *** | 3.250 *** | 3.491 *** | 5.528 *** |
| Farm type [6] |  |  |  | 0.001 ns | 1.058 ns | 2.830 ns | 0.022 ns | 2.170 ns | 1.744 ns |
| Village × Farm type [6] |  |  |  | 3.343 *** | 0.596 ns | 4.196 *** | 2.485 ** | 2.446 * | 5.265 *** |

[1]: Identifiers relate to village/village groups in Figure 2, 'v' and 'g' denote villages and village groups, respectively. [2]: 'C' = conventional, 'EE' = ecological engineering; [3]: seasons 1, 2, and 3 refer to đong xuân (winter–spring, harvested around February), hè thu (harvested around June/July) and mùa thu (harvested around October); [4]: Numbers in parentheses are standard errors, ns = $p > 0.05$, * = $p \leq 0.05$, ** = $p \leq 0.01$, *** = $p \leq 0.005$, lowercase letters indicate homogenous groups; [5]: Within factor DF are seasons 2, season × village, 18, season × farm type, 2, season × village × farm type, 16, error, 444; [6]: Between factor DF are village 9, farm type, 1, and village × farm type, 8, error, 210; [7]: Separate analyses for seasons 1 and seasons 2 and 3 were conducted to include v11 and v9, respectively.

Depending on the village in question, but not on the farm type, farmers often applied more nitrogen and phosphorus during season 2 (Table 1). Ecological engineering farmers at v1, v2, v3, g4, and g7 reported using more nitrogen, at v1, v3, g4, and g7 more potassium,

and at v1, v2, v3, and g7 more phosphorus than their neighboring conventional farmers. Meanwhile, ecological engineering farmers at v11 and g12 reported using less nitrogen and at g12 less potassium and phosphorus than their conventional neighbors (see corresponding interaction terms in Table 1).

Farmers reported planting 18 rice varieties of which OM5451 and IR50404 were grown in at least one season by >30% of farmers and Jasmine 85 and OM6976 by >10% of farmers (Table S7). The preferred varieties varied considerably between villages and, in some villages, were different between farm types. However, there were no consistent preferences for rice varieties between ecological engineering and conventional farmers across villages. A high proportion of farmers at v6, v8, and v10 rotated varieties between seasons, but this was not different between ecological engineering and conventional farmers (Table S8).

### 3.4. Pest, Disease, and Weed Management

Farmers made an average of 1.2 herbicide applications and 2.9 fungicide applications per season regardless of farm type (Figure 4). Farmers at v1 made more herbicide applications than farmers at all other villages ($p < 0.001$) (Table 2). Farmers at v10 tended to make more fungicide applications compared to farmers at all other villages except v3 and v6 ($p < 0.001$) (Table 2). Farmers made similar numbers of fungicide applications regardless of season or farm type (Table 2).

**Table 2.** Pest management actions by ecological engineering and conventional rice farmers at 12 villages/village groups in the MDR.

| Village [1] | Farm Type [2] | Season [3] | Herbicide Applications (Number) [4] | Molluscicide Applications (Number) [4] | Fungicide Applications (Number) [4] | Insecticide Applications (Number) [4] | Time of First Insecticide Application (Days) [4] | Pesticide Applications (Number) [4] |
|---|---|---|---|---|---|---|---|---|
| v1 | C | 1 | 1.11 (0.20) [a] | 0.89 (0.20) | 2.00 (0.60) [a] | 1.56 (0.56) [ab] | 26.00 (5.10) [ab] | 5.56 (1.17) [a] |
| | | 2 | 1.11 (0.20) | 0.89 (0.20) | 2.67 (0.41) | 2.33 (0.47) | 28.75 (5.07) | 7.00 (1.82) |
| | | 3 | 1.11 (0.20) | 1.00 (0.17) | 2.33 (0.50) | 2.11 (0.48) | 26.67 (4.22) | 6.56 (0.99) |
| | EE | 1 | 1.50 (0.12) | 1.33 (0.11) | 2.72 (0.32) | 1.72 (0.32) | 40.00 (2.04) | 7.28 (1.52) |
| | | 2 | 1.50 (0.12) | 1.33 (0.16) | 2.78 (0.33) | 1.94 (0.27) | 37.15 (2.54) | 7.56 (1.53) |
| | | 3 | 1.50 (0.12) | 1.22 (0.13) | 2.61 (0.33) | 1.67 (0.30) | 42.31 (2.87) | 7.00 (1.54) |
| v2 | C | 1 | 2.20 (0.20) [b] | 0.80 (0.25) | 2.50 (0.22) [a] | 2.00 (0.45) [ab] | 26.38 (5.49) [ab] | 6.82 (1.93) [a] |
| | | 2 | 2.10 (0.18) | 0.60 (0.27) | 2.50 (0.22) | 2.10 (0.43) | 28.25 (5.50) | 6.64 (1.90) |
| | | 3 | 2.20 (0.20) | 0.80 (0.25) | 2.50 (0.22) | 1.80 (0.44) | 27.63 (5.42) | 6.64 (1.90) |
| | EE | 1 | 2.00 (0.00) | 1.00 (0.00) | 3.50 (0.50) | 1.50 (1.50) | 40.00 (0.00) | 8.00 (1.00) |
| | | 2 | 1.33 (0.67) | 0.67 (0.33) | 2.33 (0.20) | 1.33 (0.88) | 40.00 (0.00) | 5.67 (1.85) |
| | | 3 | 2.00 (0.00) | 1.00 (0.00) | 3.50 (0.50) | 1.50 (1.50) | 40.00 (0.00) | 8.00 (1.00) |
| v3 | C | 1 | 1.25 (0.25) [a] | 1.00 (0.00) | 4.50 (0.29) [ab] | 1.75 (0.25) [ab] | 35.00 (4.56) [ab] | 8.50 (1.29) [a] |
| | | 2 | 1.25 (0.25) | 1.00 (0.00) | 4.50 (0.29) | 1.25 (0.75) | 40.00 (5.00) | 8.00 (1.71) |
| | | 3 | 1.25 (0.25) | 1.00 (0.00) | 4.50 (0.29) | 1.00 (0.58) | 42.50 (2.50) | 7.75 (1.48) |
| | EE | 1 | 1.20 (0.13) | 0.90 (0.18) | 3.30 (0.37) | 2.20 (0.47) | 28.14 (4.62) | 7.60 (1.69) |
| | | 2 | 1.18 (0.12) | 1.00 (0.00) | 3.36 (0.34) | 2.27 (0.49) | 25.71 (4.00) | 7.82 (1.66) |
| | | 3 | 1.20 (0.13) | 1.00 (0.00) | 3.30 (0.37) | 2.10 (0.50) | 25.83 (4.73) | 7.60 (1.65) |
| g4 | C | 1 | 1.06 (0.04) [a] | 0.90 (0.08) | 3.81 (0.13) [a] | 1.74 (0.12) [a] | 34.60 (1.08) [b] | 7.52 (1.18) [a] |
| | | 2 | 1.06 (0.04) | 0.80 (0.10) | 3.74 (0.11) | 1.77 (0.12) | 35.00 (1.11) | 7.35 (1.19) |
| | | 3 | 1.16 (0.07) | 0.84 (0.09) | 3.77 (0.12) | 1.68 (0.13) | 35.21 (1.26) | 7.45 (1.23) |
| | EE | 1 | 1.10 (0.06) | 1.66 (0.12) | 3.10 (0.20) | 0.90 (0.18) | 41.37 (2.48) | 6.76 (1.38) |
| | | 2 | 1.10 (0.06) | 1.66 (0.12) | 3.07 (0.20) | 0.90 (0.14) | 40.80 (2.42) | 6.72 (1.34) |
| | | 3 | 1.14 (0.07) | 1.52 (0.15) | 3.00 (0.17) | 0.69 (0.12) | 44.29 (1.62) | 6.34 (1.24) |
| v5 | C | 1 | 1.13 (0.09) [a] | 1.07 (0.07) | 3.67 (0.23) [a] | 3.00 (0.22) [bc] | 30.43 (3.45) [ab] | 8.87 (1.40) [a] |
| | | 2 | 1.13 (0.09) | 1.07 (0.07) | 3.73 (0.25) | 3.07 (0.21) | 33.50 (2.76) | 9.00 (1.38) |
| | | 3 | 1.20 (0.11) | 1.07 (0.07) | 3.87 (0.24) | 3.20 (0.34) | 28.71 (4.43) | 9.33 (1.54) |
| | EE | 1 | 1.50 (0.22) | 1.33 (0.21) | 3.00 (0.37) | 1.33 (0.33) | 33.75 (2.39) | 6.57 (1.07) |
| | | 2 | 1.57 (0.20) | 1.43 (0.20) | 3.00 (0.31) | 1.86 (0.34) | 40.83 (3.52) | 7.67 (1.89) |
| | | 3 | 1.57 (0.20) | 1.43 (0.20) | 3.00 (0.31) | 1.43 (0.37) | 39.00 (3.32) | 7.72 (1.99) |

Table 2. *Cont.*

| Village [1] | Farm Type [2] | Season [3] | Herbicide Applications (Number) [4] | Molluscicide Applications (Number) [4] | Fungicide Applications (Number) [4] | Insecticide Applications (Number) [4] | Time of First Insecticide Application (Days) [4] | Pesticide Applications (Number) [4] |
|---|---|---|---|---|---|---|---|---|
| v6 | C | 1 | 1.17 (0.17) [a] | 1.33 (0.21) | 3.67 (0.33) [ab] | 2.17 (0.40) [ab] | 38.75 (3.75) [ab] | 8.33 (1.56) [a] |
| | | 2 | 1.17 (0.17) | 1.17 (0.17) | 3.50 (0.43) | 2.17 (0.31) | 40.00 (4.56) | 8.00 (1.73) |
| | | 3 | 1.17 (0.17) | 1.17 (0.17) | 3.67 (0.33) | 2.00 (0.45) | 38.75 (3.75) | 8.00 (1.68) |
| | EE | 1 | 1.33 (0.21) | 1.00 (0.00) | 3.83 (0.31) | 1.50 (0.43) | 39.00 (4.00) | 7.67 (1.67) |
| | | 2 | 1.33 (0.21) | 1.00 (0.00) | 3.83 (0.31) | 1.50 (0.56) | 30.40 (5.35) | 7.67 (1.67) |
| | | 3 | 1.33 (0.21) | 1.00 (0.00) | 3.83 (0.31) | 1.33 (0.42) | 36.00 (2.92) | 7.50 (1.56) |
| g7 | C | 1 | 1.54 (0.14) [a] | 1.08 (0.08) | 2.62 (0.58) [a] | 2.00 (0.47) [abc] | 18.70 (2.89) [ab] | 7.23 (1.06) [a] |
| | | 2 | 1.54 (0.14) | 1.08 (0.08) | 2.69 (0.60) | 2.08 (0.50) | 18.70 (2.89) | 7.38 (1.11) |
| | | 3 | 1.54 (0.14) | 1.08 (0.08) | 2.77 (0.61) | 2.08 (0.47) | 18.70 (2.89) | 7.46 (1.10) |
| | EE | 1 | 1.33 (0.14) | 1.00 (0.00) | 3.00 (0.46) | 2.42 (0.43) | 39.20 (5.44) | 7.75 (1.86) |
| | | 2 | 1.33 (0.14) | 1.00 (0.00) | 3.17 (0.39) | 2.50 (0.42) | 38.70 (3.62) | 8.00 (1.80) |
| | | 3 | 1.33 (0.14) | 1.00 (0.00) | 3.17 (0.39) | 2.42 (0.48) | 37.00 (4.24) | 7.92 (1.83) |
| v8 | C | 1 | 1.33 (0.13) [a] | 0.87 (0.09) | 3.33 (0.40) [a] | 2.20 (0.28) [abc] | 26.57 (3.09) [ab] | 6.11 (1.87) [a] |
| | | 2 | 1.37 (0.11) | 0.84 (0.09) | 3.42 (0.34) | 2.16 (0.28) | 29.06 (3.03) | 7.79 (1.56) |
| | | 3 | 1.37 (0.11) | 0.89 (0.07) | 3.47 (0.32) | 1.95 (0.26) | 29.28 (3.24) | 7.68 (1.51) |
| | EE | 1 | 1.60 (0.16) | 1.22 (0.15) | 3.40 (0.27) | 2.00 (0.21) | 37.50 (2.67) | 7.36 (1.81) |
| | | 2 | 1.55 (0.16) | 1.27 (0.14) | 3.45 (0.25) | 2.00 (0.30) | 36.70 (3.65) | 8.27 (1.41) |
| | | 3 | 1.55 (0.16) | 1.27 (0.14) | 3.45 (0.25) | 1.64 (0.28) | 38.33 (2.20) | 7.91 (1.44) |
| v9 | C | 1 | - | - | - | - | - | - |
| | | 2 | 1.05 (0.05) [a] | 0.68 (0.13) | 4.05 (0.09) [a] | 1.68 (0.17) [a] | 36.67 (1.26) [b] | 7.47 (1.26) [a] |
| | | 3 | 1.11 (0.07) | 0.74 (0.13) | 4.11 (0.11) | 1.53 (0.18) | 37.50 (1.36) | 7.47 (1.31) |
| | EE | 1 | - | - | - | - | - | - |
| | | 2 | 1.55 (0.16) | 1.18 (0.12) | 3.36 (0.28) | 1.73 (0.30) | 37.70 (4.02) | 7.82 (1.46) |
| | | 3 | 1.55 (0.16) | 1.18 (0.12) | 3.36 (0.28) | 1.27 (0.27) | 38.13 (2.49) | 7.36 (1.47) |
| v10 | C | 1 | 1.38 (0.18) [a] | 1.13 (0.23) | 5.00 (0.57) [b] | 4.75 (0.56) [d] | 26.29 (3.21) [a] | 12.25 (1.00) [b] |
| | | 2 | 1.43 (0.13) | 0.76 (0.15) | 5.00 (0.39) | 4.29 (0.37) | 26.59 (1.84) | 11.48 (1.71) |
| | | 3 | 1.24 (0.14) | 0.48 (0.13) | 4.90 (0.40) | 4.29 (0.38) | 26.82 (1.79) | 10.90 (1.78) |
| | EE | 1 | - | - | - | - | - | - |
| | | 2 | 1.56 (0.18) | 0.89 (0.11) | 3.89 (0.75) | 3.11 (0.68) | 28.50 (3.23) | 9.44 (1.30) |
| | | 3 | 1.78 (0.15) | 0.56 (0.18) | 3.56 (0.65) | 3.11 (0.68) | 31.25 (3.93) | 9.00 (1.13) |
| v11 | C | 1 | 0.00 (0.00) [a] | 0.00 (0.00) | 3.80 (0.37) [a] | 2.40 (0.75) [ab] | 35.00 (0.00) [b] | 6.20 (1.07) [a] |
| | | 2 | - | - | - | - | - | - |
| | | 3 | - | - | - | - | - | - |
| | EE | 1 | 0.20 (0.20) | 0.00 (0.00) | 4.60 (0.03) | 2.20 (0.86) | 38.33 (3.33) | 7.00 (1.00) |
| | | 2 | - | - | - | - | - | - |
| | | 3 | - | - | - | - | - | - |
| g12 | C | 1 | 1.35 (0.11) [a] | 1.00 (0.07) | 3.35 (0.28) [a] | 3.00 (0.19) [c] | 30.53 (1.62) [ab] | 8.70 (1.45) [a] |
| | | 2 | 1.40 (0.11) | 0.95 (0.09) | 3.35 (0.31) | 2.95 (0.27) | 30.79 (1.84) | 8.65 (1.45) |
| | | 3 | 1.35 (0.11) | 1.00 (0.07) | 3.00 (0.26) | 3.00 (0.23) | 29.05 (1.36) | 8.35 (1.44) |
| | EE | 1 | 1.10 (0.10) | 1.00 (0.00) | 3.50 (0.34) | 3.40 (0.27) | 27.50 (2.27) | 9.00 (1.52) |
| | | 2 | 1.20 (0.13) | 1.00 (0.00) | 3.60 (0.34) | 3.50 (0.27) | 29.50 (2.83) | 9.30 (1.58) |
| | | 3 | 1.10 (0.10) | 1.00 (0.00) | 3.60 (0.43) | 3.70 (0.30) | 28.50 (2.59) | 9.40 (1.69) |
| Season [5,7] | | | 0.748 [ns] | 4.129 [*] | 1.017 [ns] | 3.694 [*] | 1.153 [ns] | 2.353 [ns] |
| Season × Village [5,7] | | | 1.338 [ns] | 1.995 [**] | 1.302 [ns] | 1.373 [ns] | 0.933 [ns] | 2.311 [***] |
| Season × Farm type [5,7] | | | 0.686 [ns] | 1.569 [ns] | 0.189 [ns] | 0.242 [ns] | 1.487 [ns] | 0.453 [ns] |
| Season × Village × Farm type [5,7] | | | 0.213 [ns] | 0.431 [ns] | 1.175 [ns] | 0.901 [ns] | 0.953 [ns] | 1.173 [ns] |
| Village [6] | | | 4.387 [***] | 1.347 [ns] | 3.705 [***] | 11.690 [***] | 3.896 [***] | 5.255 [***] |
| Farm type [6] | | | 0.651 [ns] | 6.258 [**] | 0.034 [ns] | 2.219 [ns] | 7.923 [***] | 0.003 [ns] |
| Village × Farm type [6] | | | 1.578 [ns] | 3.393 [***] | 1.563 [ns] | 2.691 [**] | 3.878 [***] | 1.236 [ns] |

[1]: Identifiers relate to village/village groups in Figure 2, 'v' and 'g' denote villages and village groups, respectively. [2]: 'C' = conventional, 'EE' = ecological engineering; [3]: seasons 1, 2 and 3 refer to đong xuân (winter–spring, harvested around February), hè thu (harvested around June/July) and mùa thu (harvested around October); [4]: Numbers in parentheses are standard errors, ns = $p > 0.05$, * = $p \leq 0.05$, ** = $p \leq 0.01$, *** = $p \leq 0.005$, lowercase letters indicate homogenous groups; [5]: Within factor DF are seasons 2, season × village, 18, season × farm type, 2, season × village × farm type, 16, error, 444; [6]: Between factor DF are village 9, farm type, 1, and village × farm type, 8, error, 210; [7]: Separate analyses for seasons 1 and seasons 2 and 3 were conducted to include v11 and v9, respectively.

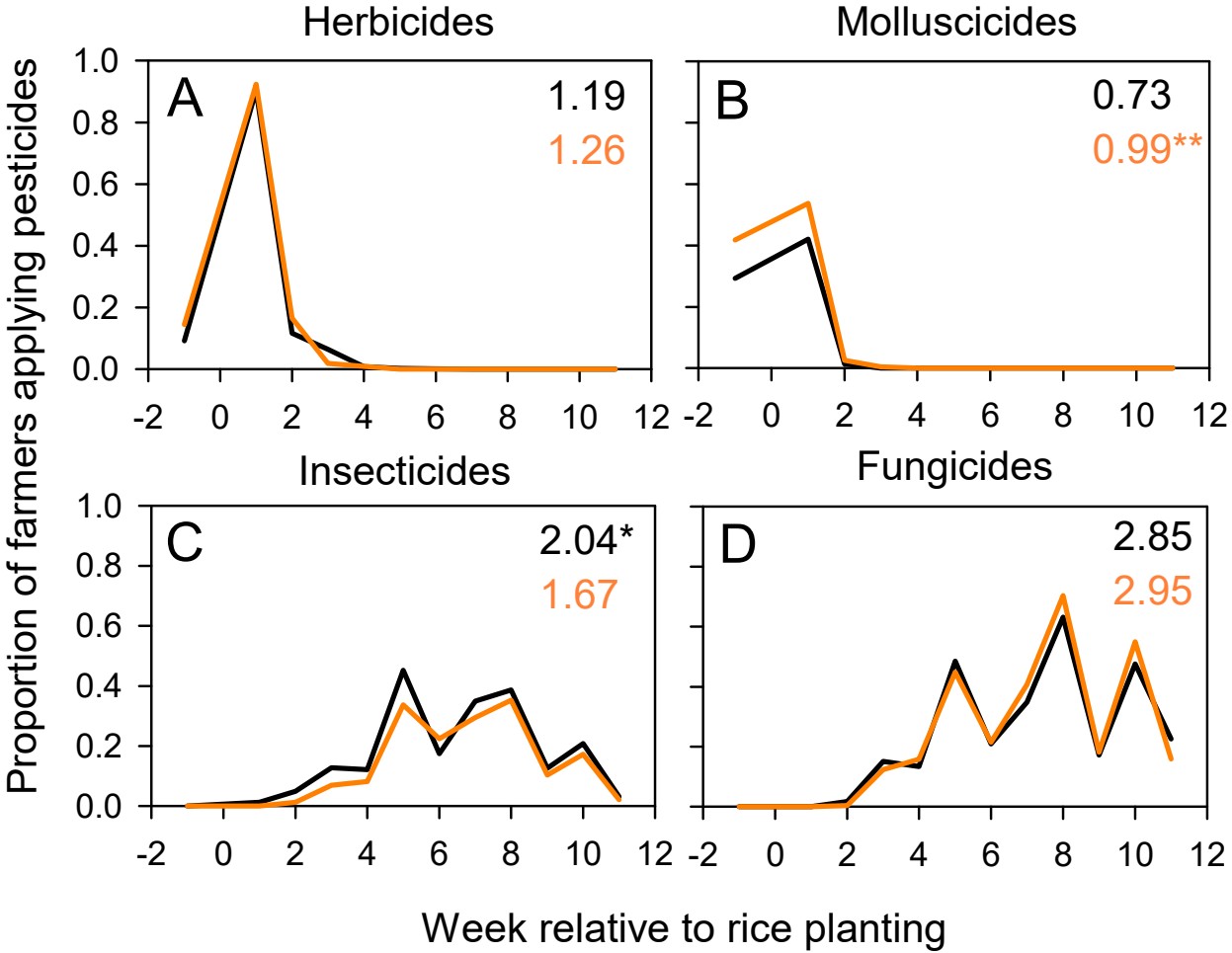

**Figure 4.** Weekly applications of (**A**) herbicides, (**B**) molluscicides, (**C**) insecticides, and (**D**) fungicides as reported by ecological engineering (orange lines) and conventional (black lines) farmers at 12 villages/village groups during an average rice crop (i.e., averaged across three seasons). Numbers indicate total applications of each pesticide type per farmer, per season. Significantly higher numbers of applications are indicated as *, $p \leq 0.05$, and **, $p \leq 0.01$. Further details of total applications are presented in Table 2.

Ecological engineering farmers tended to make more molluscicide applications at villages v1, v2, g4, v5, v8, and v9 but not at the other villages (interaction, $p < 0.001$) and farmers applied more molluscicides during seasons 1 and 3 ($p < 0.05$) (Table 2). Herbicides and molluscicides were applied from one week before planting and up to week 2. Fungicides were mainly applied at weeks 3, 5, 8, and 10 (Figure 4).

Although many farmers rotated rice with other crops (Table S8), none of the farmers rotated crops for pest or disease management purposes. Instead, rotations were due to changing crop values and based on water availability (see also Horgan et al. (2022) [14]). Over 84% of farmers thought that their preferred varieties had some insect or disease resistance; however, few farmers could specifically report resistance: 70% of farmers that grew OM5451 thought the variety was resistant to blast disease (*Magnaporthe grisea* (T.T. Hebert) M.E. Barr), ca 26% resistant to the brown planthopper, and ca 14% resistant to leaffolders (*Cnaphalocrocis medinalis* (Guenée) and/or *Marasmia patnalis* Bradley). Over 44% of farmers that grew OM6976 thought it was resistant to the brown planthopper (Table S7). Perceptions of varietal resistance varied between villages; for example, farmers at v8, v9, v10, and g12 thought that OM5451 was resistant to brown planthopper, whereas, at v1 and g4, farmers thought the same variety was resistant to leaffolders. A small number of farmers thought their varieties were resistant to other pests and diseases (Table S7).

Ecological engineering farmers at v1, v2, g4, v5, and v10 made fewer insecticide applications and began applications later than their conventional neighbors; ecological engineering farmers at v6 also made fewer applications, and farmers at g7 and v8 began applications later. However, ecological engineering farmers at v3, g7, and g12 made more insecticide applications and at v3 and g12 they also applied their insecticides earlier than their conventional farming neighbors (interactions, number of applications, $p < 0.01$; timing of first application, $p < 0.001$) (Table 2). Insecticides were generally applied during weeks 3, 5, 8, and 10 after planting (Figure 4).

PERMANOVA results showed significant differences between villages in the choice of insecticide used (F = 5.131, $p < 0.001$), but not between ecological engineering and conventional farmers (F = 1.853, $p = 0.056$). However, there was a significant interaction between both variables (F = 1.628, $p < 0.001$) because differences in insecticide use based on farm type were apparent at g4 and v9 ($p \leq 0.001$), but not at the other villages (all $p > 0.05$) (Figure 5). SIMPER test results indicate similar patterns in insecticide choice based on farm type at these two villages. In particular, fipronil, quinalphos, and chlorantraniliprole + thiamethoxam were the most commonly used insecticides by conventional farmers, whereas flubendiamide was most commonly used by ecological engineering farmers (Figure 5).

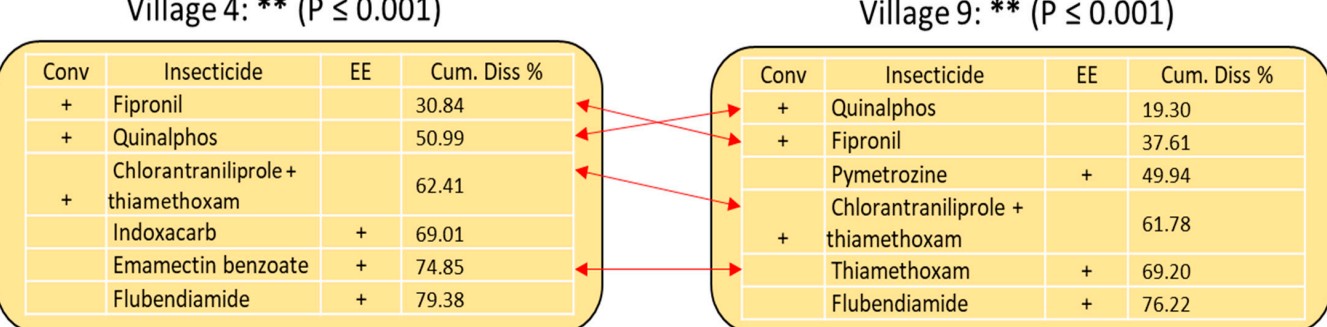

**Figure 5.** Differences between farmers' reported insecticide use based on rice management (conventional, 'Conv' or ecological engineering, 'EE'). Results of PERMANOVA significant pair-wise analyses are shown, ** = $p \leq 0.01$. SIMPER analyses (in orange boxes) show the insecticides contributing most to the dissimilarity between groups. The '+' symbols indicate the group in which each insecticide was more reported. 'Cum. Diss %' = cumulative dissimilarity between groups expressed as a percentage. Arrows highlight similar patterns between villages.

Overall, there were no differences in the total number of pesticide applications by ecological engineering and conventional farmers in any season because farmers with lower insecticide inputs tended to increase their applications of molluscicides and/or fungicides (Figure 4, Table 2). Farmers at v10 made more pesticide applications compared to farmers at the other villages, irrespective of farm type (Table 2).

*3.5. Predictors of Pesticide Use*

The numbers of pesticide (i.e., insecticide, herbicide, fungicide, molluscicide, and total pesticide) applications that farmers made were heavily influenced by 'village' (Table 3). 'Variety' was a prominent predictor of pesticide use in all models, but its contribution varied between models, being higher in all models (explained variance $\geq 4.87\%$) other than in the molluscicide-use model (explained variance = 1.66%). 'Farm type' explained little of the variation in most models except in the molluscicide-use model, in which it was the second predicter variable (explained variance = 7.71%). The herbicide use model was the best fitted model (i.e., lowest AICc value), explaining 33.1% of the variation in the data based on four variables: 'village', 'variety', 'varietal resistance to pests', and 'farm type' (Table 3).

**Table 3.** Distance-based linear models summary, with predictor variables included in each of the models.

| Dependent Variable | Model Summary | | | Sequential Tests | | | | |
|---|---|---|---|---|---|---|---|---|
| | AICc | $R^2$ | No. of Predictors | Predictor Variable | Pseudo-F | *p*-Value | % Var. expl [1] | % Cum. Var [2] |
| Insecticide use | 3584.9 | 0.226 | 6 | Village | 44,813 | 0.002 | 10.94 | 10.94 |
| | | | | Variety | 46,817 | 0.002 | 5.57 | 16.51 |
| | | | | Phosphorus | 28,162 | 0.002 | 2.7 | 19.21 |
| | | | | Farm type | 17,930 | 0.002 | 1.68 | 20.89 |
| | | | | Nitrogen | 11,228 | 0.003 | 1.02 | 21.91 |
| | | | | Variety Res. Pest | 7632 | 0.008 | 0.68 | 22.59 |
| Herbicide use | 2345.5 | 0.331 | 4 | Village | 74,909 | 0.002 | 26.83 | 26.83 |
| | | | | Variety | 22,128 | 0.002 | 4.99 | 31.82 |
| | | | | Variety Res. Pest | 8711 | 0.010 | 0.79 | 32.61 |
| | | | | Farm type | 5117 | 0.0259 | 0.46 | 33.07 |
| Molluscicide use | 3016.8 | 0.234 | 6 | Village | 66,759 | 0.002 | 9.84 | 9.84 |
| | | | | Farm type | 64,939 | 0.002 | 7.71 | 17.55 |
| | | | | Potassium | 20,882 | 0.002 | 2.22 | 19.77 |
| | | | | Variety | 15,838 | 0.002 | 1.66 | 21.43 |
| | | | | Education | 5793 | 0.014 | 1.06 | 22.49 |
| | | | | Farming experience | 4932 | 0.029 | 0.48 | 22.97 |
| | | | | Age | 4144 | 0.048 | 0.41 | 23.38 |
| Fungicide use | 3174.7 | 0.242 | 9 | Village | 42,965 | 0.002 | 8.12 | 8.12 |
| | | | | Variety | 37,696 | 0.002 | 7.11 | 15.23 |
| | | | | Education | 27,317 | 0.002 | 3.07 | 18.3 |
| | | | | Nitrogen | 16,771 | 0.002 | 1.76 | 20.06 |
| | | | | Age | 12,930 | 0.002 | 1.41 | 21.47 |
| | | | | Farm type | 8288 | 0.004 | 0.84 | 22.31 |
| | | | | Phosphorous | 7234 | 0.008 | 0.77 | 23.08 |
| | | | | Type of planting | 5712 | 0.02 | 0.57 | 23.65 |
| | | | | Rice area | 5570 | 0.014 | 0.56 | 24.21 |
| Pesticide use | 3058.8 | 0.233 | 6 | Village | 39,579 | 0.002 | 12.07 | 12.07 |
| | | | | Variety | 19,577 | 0.002 | 4.87 | 16.94 |
| | | | | Phosphorous | 18,577 | 0.002 | 2.16 | 19.1 |
| | | | | Education | 15,933 | 0.004 | 1.72 | 20.82 |
| | | | | Age | 15,836 | 0.002 | 1.68 | 22.5 |
| | | | | Nitrogen | 7541 | 0.010 | 0.77 | 23.27 |
| Yield | 3080.2 | 0.231 | 6 | Type of planting | 65,227 | 0.002 | 11.97 | 11.97 |
| | | | | Variety | 59,718 | 0.002 | 4.87 | 16.84 |
| | | | | Village | 12,836 | 0.002 | 4.54 | 21.38 |
| | | | | Fallow | 6867 | 0.012 | 0.67 | 22.05 |
| | | | | Farm type | 5792 | 0.014 | 0.56 | 22.61 |
| | | | | Phosphorous | 4664 | 0.029 | 0.45 | 23.06 |

[1] explained variance; [2] cumulative variance.

### 3.6. Predictors of Reported Rice Yields

Farmers reported higher yields for season 2; however, there was a significant season × village interaction because of generally higher yields at v1 and g7 during season 2 than at the remaining villages ($p < 0.05$) (Table 1). Ecological engineering farmers at v2 reported higher yields than their conventional neighbors, but yields were similar between farm types for the remaining villages (interaction, $p < 0.001$) (Table 1). Crop establishment method and variety were the best predictors of yields, with 'village' included as the third predictor variable—together explaining over 21% of the variance (Table 3).

### 3.7. Pest and Disease Concerns

Farmers reported a range of rice pest and disease problems each season. Blast was the main biotic constraint mentioned by farmers (mentioned by 19–56% as their principal problem each season: Table S9). Leaffolders (15–21%), planthoppers (2–12%), and panicle mite (*Steneotarsonemus spinki* Smiley) (1–8%) were regarded as the most problematic arthropod pests (Table S9).

Farmers' perceptions of the most harmful pests and diseases differed significantly between villages (F = 8.528, $p$ < 0.001), but not between farm types (F = 2.036, $p$ = 0.06). However, there was a significant village × farm type interaction, since farmers' perceptions changed based on farm type at g4, v8, and v9, but not at the other villages (F = 2.765, $p$ = 0.001) (Figure 6). Based on SIMPER test results, blast was reported as the most harmful disease by conventional farmers in all three villages, whereas leaffolders were the most common harmful pest reported by ecological engineering farmers (Figure 6).

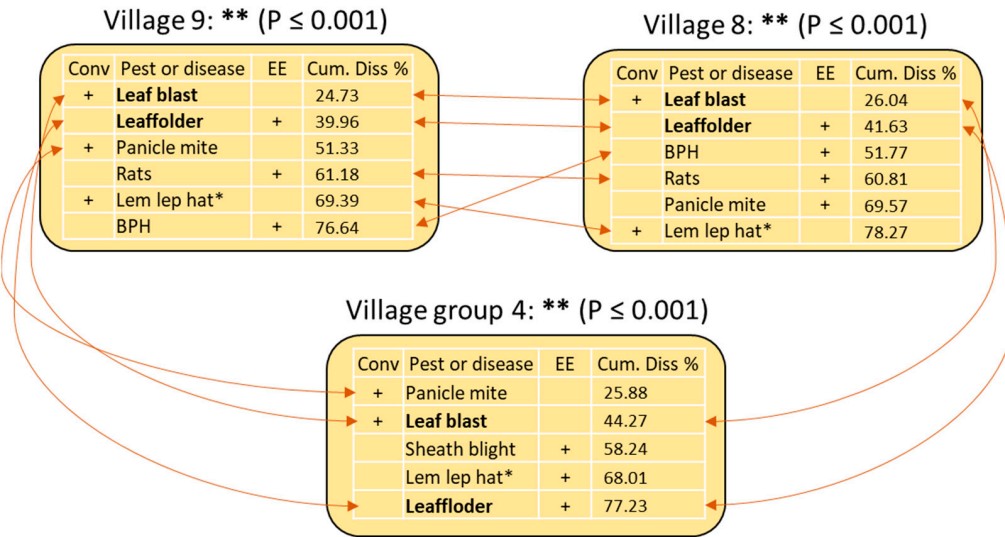

**Figure 6.** Differences between farmers' perceptions of the most harmful pests and diseases based on rice management (conventional, 'Conv' or ecological engineering, 'EE'). Results of PERMANOVA significant pair-wise analyses are shown, ** = $p \leq 0.01$. SIMPER analyses (in orange boxes) show the pests and diseases contributing most to the dissimilarity between groups. The '+' symbols indicate the farm type in which each pest or disease was more reported. 'Cum. Diss %' = cumulative dissimilarity between groups expressed as a percentage. Arrows highlight similar patterns between villages, and bold letters indicate coincidences between the three villages. Note: Lem lep hat* refers to unfilled grains caused by diseases, as reported by farmers. See Table S9 for further details related to perceived pests and diseases, including scientific names.

## 4. Discussion

Based on the results from our survey, village membership was often the principal determinant of farmers' agronomic responses to flower strips in the MDR (Tables 1–3); thereby implying that communication between stakeholders—including farmers, village leaders, agricultural extensionists, and agrochemical dealers—plays a significant role in determining agronomic and pest management responses to community-based agroecological interventions. In the following sections, we assign ecological engineering farmers from each of the 12 villages/village groups to one or more of six models for their integration of flower strips and pest management approaches based on the indicators outlined in Figure 1. We suggest some factors that may have influenced integration methods and make recommendations to gain extra benefits from ecological engineering and further reduce the often-considerable pesticide use reported by farmers.

### 4.1. Flower Strips and Insect Management Options

It is reasonable to predict that ecological engineering would have eliminated insecticide use on participating rice farms either because farmers wished to restore ecosystem services or because these farmers employ IPM principles by adhering to threshold-based curative actions (models 1 and 2, Figure 1)—with action thresholds rarely being met. However, when we distinguished between IPM and prophylactic farmers based on few and irregular insecticide applications (IPM) or many and regular applications (prophylactic), we found

that only 16 of 305 farmers likely practiced IPM. Twelve of these farmers were ecological engineering farmers, and most were associated with village v1. None of the farmers we interviewed practiced organic farming, and only one used no pesticides (farming rice in only one season). Nineteen farmers used no insecticides, of which twelve were ecological engineering farmers, mainly at village g4—we do not know if these insecticide-free farmers also practiced IPM by monitoring their fields for potential pest-related yield losses. Therefore, the basic model of integration (model 1) and the insecticide-free model (model 2) were apparent to some extent at v1 and g4; but this represented only 25 (18.8%) of the 133 farmers that planted flower strips. All other ecological engineering farmers continued to use insecticides and other pesticides prophylactically.

Models 3 and 4 suggest that farmers might have focused specifically on reducing pesticides during early rice crop stages by viewing flower strips as a replacement for one or more early ($\leq$40 days) insecticide applications or because they believed that the flower strips only functioned to enhance regulation during the early crop (Figure 1). For more than 20 years prior to initiating ecological engineering in the MDR and at the same time as the ecological engineering movement was promoted, national extension programs had advised rice farmers to avoid spraying insecticides before 40 days to allow natural enemies to build up numbers and, thereby, better regulate pest populations [8,41–43]. The best known of these programs was the 3R3G program initiated in 2003, which advocated reducing fertilizer levels, seeding rates, and insecticide applications (three reductions) [9,42]. A further, derived, program was the 'One Must Do, 5 Reductions' ('Mot Phai, Nam Giam'—1M5R) initiated in 2010, which further advocated reducing water consumption and post-harvest losses together with the necessary use of certified rice seed (the 'must do') [9]. Agricultural extension officers would have been widely familiar with these programs at the time that ecological engineering was promoted. It was therefore likely that farmers who planted flower strips might also be more compliant with the 3R3G and other programs. Our results suggest that this was the case at a number of villages: more ecological engineering farmers at g4, v5, and g7 avoided insecticides before 40 days compared to their conventional neighbors. At g7, farmers tended to shift an early application to a later crop stage—that is, they made the same or more insecticide applications compared to their conventional neighbors—but on average made their applications at later crop stages (Table 2).

Ecological engineering farmers at v6 and g12 made fewer insecticide applications compared to their conventional neighbors while still making applications before 40 days. This suggests that these farmers largely followed model 4, whereby flower strips replaced some other pest management action—in most cases, a single insecticide application at a later crop stage (Figure 1). Because the 1M5G program and ecological engineering were promoted through the same authorities and during the same years, it is difficult to determine which program ultimately convinced farmers to reduce insecticide use; however, it is apparent that many farmers at some villages were largely unwilling to reduce insecticide inputs without some alternative pest management option such as establishing flower strips. Proponents of 1M5G have already recognized parallel programs (such as the Vietnam Good Agricultural Practices and the Rice Sustainability Platform) as influencing adoption success by establishing complementary agronomic and pest management targets [9]; however, the role of ecological engineering, which, by 2015, was already practiced by >13,000 households on >14,000 ha of rice in the MDR (personal communications with the Southern Regional Plant Protection Centre), in contributing to sustainable rice crop management may have been overlooked in determining some of the successes of these programs. We suggest that the flower strips functioned largely to alleviate farmers' fears of pest-related crop losses when insecticides were avoided at early crop stages thereby facilitating the adoption of 3R3G, IM5R, and other sustainability practices among some farmers. Across the villages that we surveyed, almost 30% more ecological engineering farmers (i.e., 12 farmers) than conventional farmers avoided using pesticides until after 40 days (i.e., 38% and 27%, respectively, which is equal to 9% of ecological engineering farmers excluding insecticide-free and IPM farmers).

Our model 4 also included the possibility that rice farmers might reduce some other pest management options through replacement with flower strips. In general, the only other pest management practice that farmers mentioned was the planting of resistant rice varieties. However, when we checked against records for the most commonly planted varieties, we found no published evidence of any pest resistance among the varieties that the farmers planted [57–59]. Meanwhile, ecological engineering farmers at v8, v9, and v11 made similar numbers of insecticide applications compared to their conventional neighbors—suggesting that these farmers either planted their flower strips for some other reason (i.e., to sell flowers) or had little understanding of the role of the strips in pest management (model 5, Figure 1). Many ecological engineering farmers at v2, v3, g7, and g12 made more insecticide applications compared to their neighbors, thereby suggesting that these farmers may have seen the flower strips as an '*insurance*' against pest outbreaks, in effect, allowing the farmers to be less discriminate with their applications. This type of response has been reported previously among farmers who planted brown planthopper-resistant rice in Indonesia [60]. Overall, our multivariate analysis of insecticide responses by farmers suggests that the flower strips played only a minor role (explaining 1.68% of the variance, Table 3) in determining the number of insecticide applications by rice farmers. Rice variety was a better predictor of applications, with BTE-1 and Jasmine 85 generally receiving higher numbers of insecticide applications. BTE-1 is a hybrid variety and Jasmine 85 is an aromatic rice—both varieties are notably susceptible to pests and diseases [61].

We detected differences between ecological engineering and conventional farmers in their choices of pesticides at some villages/village groups. For example, conventional farmers at g4 and v9 tended to apply chlorantraniliprole + thiamethoxam, fipronil, and quinalphos more than their ecological engineering neighbors. At these same villages, ecological engineering and conventional farmers differed in the principal pest problems they reported. At g4 and v9, the main pest problem for conventional farmers was panicle mite, which likely developed in response to some of the pesticides used [62,63] and was combated using the organophosphate quinalphos. At villages g4, v8, and v9, ecological engineering farmers were most concerned with leaffolders. Damage from this pest is most apparent at later crop stages (i.e., after 40 days) when many of the farmers were already applying insecticides (Figure 4). Leaffolders have been shown to increase in densities in response to certain insecticides [19,64,65]—but further study is required to better explain why ecological engineering farmers may have considered it more problematic. At v8 and v9, rats were of more concern to ecological engineering farmers compared to conventional farmers. This may have been due to the dense vegetation on the bunds that likely provides habitat and refuge for rodents [66]. Horgan et al. (2017) [44] used vegetation patches instead of flower strips to avoid problems with rats in ecologically engineered rice following observations of increased rodent activity associated with flower strips in the Philippines. However, as indicated in the results of this study, rats are only problematic in some villages.

### 4.2. Other Factors Determining the Success of Flower Strips

Ecological restoration using flower strips or otherwise diversified habitat is expected to bring several benefits besides improving the natural regulation of arthropod pests [14,44,45]. Furthermore, as a restoration practice, farmers might have been expected to reduce their use of other pesticides because herbicides and fungicides are also hazardous to the environment and some products are associated with outbreaks of insect pests [40]. However, based on our analyses, ecological engineering farmers in the MDR only associated flower strips with reducing the need for insecticides. Ecological engineering had a minimal effect on herbicide (0.46% of variation, Table 2) or fungicide (0.84% of variation, Table 2) use, and ultimately brought no reduction in overall pesticide use (Table 2). Furthermore, ecological engineering was associated with notably higher molluscicide use (7.71% of variation, Table 2). We cannot explain based on our survey results why ecological engineering farmers use more molluscicide compared to conventional farmers. It is unlikely that terrestrial plants grown on raised bunds would have affected the densities of golden apple snails (*Pomacea canalicu-*

*lata* (Lamarck)—an aquatic rice pest). Indeed, apple snail densities are expected to decline in ecologically engineered rice fields due to greater predation by birds and rodents along vegetation bunds [44,67]. Whatever the reason for the higher molluscicide use by ecological engineering farmers, it is highly probable that the mechanisms were socio-economic and not a response to any ecological changes in the rice ecosystem (see below).

The large number of pesticide applications reported each season (often one application of some pesticide every one or two weeks) indicates that pesticides were unlikely to represent an economic constraint for the farmers. Although we did not record pesticide costs or farmer expenditure on pesticides, it is apparent that farmers were not generally constrained by pesticide costs (Table 2). A low cost of pesticides also suggests that farmers will be unlikely to apply IPM practices—because action thresholds based on the relative costs of pesticide inputs and perceived pest damage will be normally low [8,68]. This differs from results in other regions where farmers sometimes reduce pesticide use because of high costs [69]. Our multivariate analyses also indicate that pesticide use was related to crop production practices, including fertilizer applications, and sociological factors such as education, age, and village membership; thereby largely supporting previous studies from Cambodia [3,5]. Indeed, according to our results, compared to production technologies (lock-in), stakeholder communication (as indicated by the influence of village membership) greatly contributed to varying pesticide inputs (Table 3).

The high yields reported by some farmers in our study were mainly associated with machine sowing and transplanting (as opposed to manual direct seeding); Jasmine 85, Nep, OM7347, and OM5451 varieties (as opposed to C10 and OM4218 particularly); and village membership. Yields were apparently not influenced by pesticide inputs (which ranged from 0–18 applications per season) or by flower strips (Table 3), thereby reaffirming that pests have little impact on rice yields [11,18,29,41] and many pesticides (particularly insecticides) are largely unnecessary in rice production systems. This is in agreement with a report by Vo et al. (2015) [35] that ecological engineering farmers in the MDR had greater profitability because they reduced insecticide applications, but had similar yields to conventional farmers. This also draws attention to the need to carefully explain any association between pesticide use and yields at regional or national levels because, at larger spatial and temporal scales, higher yields (and greater profits) might be the cause of greater pesticide inputs and not the other way around. Other factors must also be considered when assessing the impact of flower strips on pesticide use and yields.

Although ecological engineering and conventional farmers were similar in age, education, and rice-growing experience, the ecological engineering farmers we surveyed had several socio-economic advantages—for example, ecological engineering farmers often had larger land areas and rented extra land for rice planting (Table S4); at g4, more ecological engineering farmers used rice transplanting or sowing machines; and at several villages, the ecological engineering farmers applied more nitrogen, potassium, and phosphorus, and more molluscicides. An association between socio-economic advantages and the adoption of ecological engineering by farmers may be due to targeting by extension services of the most progressive farmers [70,71] and this could have contributed to a lack of an effect in reducing pesticide inputs at certain villages where other aspects of extension (i.e., pest and nutrient management) were poorly coordinated with the promotion of flower strips [6,9,36,37]. As has been indicated in a series of previous studies, ecologically based interventions are often either hindered (functionally) by continuing pesticide use or are sometimes ineffective at reducing pesticide inputs in the face of aggressive pesticide marketing and the promotion of pesticides through agricultural extension services [4,17,50,72–74]. This suggests that flower strips alone, without the further promotion of environmental capital (e.g., the harvesting of wild or farmed fish, or the production of honey [14,46,53]), will have a limited capacity to deliver on sustainability targets unless better coordinated with ongoing and long-term training programs.

## 5. Conclusions

Ecological engineering of rice paddies in the MDR was associated with a statistically significant reduction in insecticide use and a delay in insecticide applications among participating farmers. However, ecological engineering farmers and conventional farmers made similar numbers of pesticide applications overall and in each season, partly because ecological engineering farmers applied more molluscicides and fungicides. Furthermore, many participating farmers made similar (i.e., herbicides and fungicides) or more frequent (i.e., molluscicides) applications of other pesticides compared to their conventional neighbors. This indicates that a majority of participating farmers regarded ecological engineering as a pest management option and not as a restoration practice. Furthermore, the pest management responses by farmers were greatly affected by village membership, suggesting that the outcome of ecological engineering was largely determined by related extension services and/or knowledge supply. Therefore, based on this case study from Vietnam, we suggest that greater attention to the training of farmers in ecosystem restoration is required to gain further benefits from the implementation of ecological engineering in Asian rice landscapes, and possibly for other crops also.

**Supplementary Materials:** The following supporting information can be downloaded at: https://www.mdpi.com/article/10.3390/su151612508/s1, Table S1: Sections and related questions from the questionnaire; Table S2: List of predictor variables included in the calculation of the DistLM models; Table S3: Farmer profiles at 12 villages/village groups; Table S4: Farmer land ownership and land areas at 12 villages/village groups with percentages of farmers rotating rice with other crops in some years; Table S5: PERMANOVA pair-wise test results of differences between villages in bund plant species composition; Table S6: Flowering plants grown on bunds as indicated in Figure 2; Table S7: Varieties planted by rice farmers at 12 villages/village groups with farmers' perceptions of pest and disease resistance for each variety; Table S8: The potential role of varieties in IPM according to interviewed farmers; Table S9: Biotic constraints reported by farmers for each surveyed season; Figure S1: Rice cropping at sites in the MDR of Vietnam based on farmer reported planting and harvesting dates.

**Author Contributions:** Conceptualization, F.G.H. and E.C.-M.; methodology, F.G.H. and Q.V.; formal analysis, F.G.H. and E.C.-M.; investigation, F.G.H., Q.V. and E.C.-M.; resources, F.G.H., E.A.M. and E.C.-M.; data curation, F.G.H. and Q.V.; writing—original draft preparation, F.G.H., Q.V., S.D., M.D., J.S. and E.C.-M.; writing—review and editing, F.G.H., Q.V., E.A.M., S.D., M.D., J.S. and E.C.-M.; visualization, F.G.H. and E.C.-M.; supervision, F.G.H., M.D. and J.S.; project administration, F.G.H.; funding acquisition, F.G.H., E.A.M., M.D. and J.S. All authors have read and agreed to the published version of the manuscript.

**Funding:** This research was funded by the Global Rice Science Platform (GRiSP) under the directorship of Achim Dobermann; the German Federal Ministry of Education and Research (BMBF) as part of the LEGATO project (grant number: 01LL0917A); and the Comisión Nacional de Investigación Científica y Tecnológica (CONICYT, Chile), ANID PIA SOC 180040—project Associative Research Program (Programa de Investigación Asociativa) Anillos of Social Sciences and Humanities to the University of Maule (Chile).

**Institutional Review Board Statement:** Not applicable.

**Informed Consent Statement:** Informed consent was obtained from all subjects involved in the study without obligation to answer any of the questions on the questionnaire. No record of farmer names or other information to link farmers to individual questionnaires has been maintained. Surveys were coordinated through national and local authorities to ensure the protection of those involved in the study.

**Data Availability Statement:** The data presented in this study are available on reasonable request to the corresponding author.



**Acknowledgments:** The authors thank the coordinators of field activities for their valuable support in organizing survey activities; the interviewers who conducted the interviews; and the farmers for their participation in focus group discussions, pre-testing and structured interviews. We thank Ho Van Chien, Le Quoc Cuong, and Le Thi Dieu Xuan of the Southern Regional Plant Protection Center, Tien Giang Province; To Huynh Nhu of the Plant Protection Department (PPD) of Vung Liem District, Vinh Long Province; Le Hieu and Nguyen Thi Ai M, PPD, Bac Lieu Province; Tran Van Duong, PPD, An Giang Province, and Nguyen Thi Hau, PPD, Kien Giang Province. We are grateful to the entomology staff at the International Rice Research Institute, particularly Maria Liberty P. Almazan, Carmencita Bernal, and Angelee Fame Ramal, for helpful discussions on aspects of the survey and Michael Eddleston for comments on the manuscript.

**Conflicts of Interest:** The authors declare no conflict of interest.

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
