# Peer review of "Escaping the Lock-in to Pesticide Use: Do Vietnamese Farmers Respond to Flower Strips as a Restoration Practice or Pest Management Action?"

_sustainability, doi:10.3390/su151612508_

Round 1

Reviewer 1 Report

The manuscript, “Escaping the lock-in to pesticide use: Do Vietnamese farmers respond to flower strips as a restoration practice or pest management action?”, by Finbarr G. Horgan et al. assess the agronomic and pest management responses by MDR rice farmers to flower strips and established that ecological engineering is best viewed as a holistic approach to ecosystem restoration (that includes pest regulation services) if farmers are to escape the lock-in to pesticide use resulting from intensification and from the availability and marketing of pesticides. The manuscript is acceptable after minor changes. My comments have been given below:

(1)  The introduction could be elaborated with some more recent literature reports.

(2)  Abstract is too lengthy; it should be reduced to some extent.

(3)  The formatting of the reference section is required.

Author Response

We thank the reviewer for his/her attention to the manuscript and valuable suggestions. We have now responded to all his/her suggestions as follows:

(1)  The introduction could be elaborated with some more recent literature reports. - –we have added text and several references to give a better overall picture of recent advances in pest management.

(2)  Abstract is too lengthy; it should be reduced to some extent. – we have now reduced the abstract to 225 words while maintaining most of the original information.

(3)  The formatting of the reference section is required. – we have now corrected those references that were improperly formatted.

Reviewer 2 Report

Dear Authors

This manuscript provides a nice study in Escaping the lock-in to pesticide use: Do Vietnamese farmers respond to flower strips as a restoration practice or pest management action?. The topic is original in the field. The conclusions consistent with the evidence and arguments presented and they address the main question posed. The tables and figures are ok.

However, the reviewer thinks there are some aspects that can be improved answered as listed below:

1. The introduction could be improved and more focused on new researches in this field. 

2. This reviewer thinks that an additional section in relation to novel aspects of this work.

3. Please write the aims of this research, clearly.

4. Add these new papers in introduction or discussion sections:

Encapsulation of Plant Biocontrol Bacteria with Alginate as a Main Polymer Material

A novel encapsulation of Streptomyces fulvissimus Uts22 by spray drying and its biocontrol efficiency against Gaeumannomyces graminis, the causal agent of take-all disease in wheat

Finally, after these minor corrections, this manuscript can be publish. 

Author Response

We thank the reviewer for his/her valuable comments. We have made all changes as requested and indicated below:

  1. The introduction could be improved and more focused on new researches in this field. – in response to the reviewer’s comment, we have added text and several references including references related to those proposed by the reviewer (below) to give a better overall picture of recent advances in pest management (lines 80-85, 1102-1124). This cannot be too exhaustive, since we are discussing a very specific type of preventative action, whereas the reviewer seems to be seeking (based on indicated references) more information on advances in curative actions.
  2. This reviewer thinks that an additional section in relation to novel aspects of this work. – because another reviewer has asked that the text be reduced, we have not included a new section on novel aspects of the work, but rather included these aspects as additional sentences in the introduction and discussion (lines 141-145; 996-999).
  3. Please write the aims of this research, clearly. – we have further elaborated on the aims of the study as requested by the reviewer. These changes are indicated at the end of the introduction (lines 133-140).
  4. Add these new papers (listed by the reviewer) in introduction or discussion sections: - Please see response as part of comment 1 above. One of the papers seems to describe a possible product for development that is not yet available or tested in the field – we therefore did not include that particular reference; however, we did include a review by the same author.

Reviewer 3 Report

Manuscript is original, well defined, easy to understand. The language is appropriate and understandable. The topic is compatible with the journal’s scope. The results are very significant and relevant, presented in a well-structured manner. The manuscript’s results are reproducible based on the details given in the methods section. The figures (1-6) and tables (1-3) are appropriate, they are clearly presented. Conclusions justified and supported by the results, consistent with the evidence and arguments presented. References are listed according to the regulations of the publishers.

The key contribution of this manuscript is the impact on agricultural policy that should aim to reduce the use of pesticides and prevent environmental and health consequences. It is necessary to continue with the further development of a strategy that will help farmers overcome the pesticide blockade.

Accept manuscript with minor changes:

Line 42: Add a reference/references

Line 194: Place the Latin name of the species in parentheses after the ordinari name

Line 195: Same as line 194

Author Response

We thank the reviewer for his/her kind comments on the manuscript and his/her suggestions for changes. We have responded to these requests as follows:

Line 42: Add a reference/references – references have now been indicated to back up the statement.

Line 194: Place the Latin name of the species in parentheses after the ordinary name – corrected here and throughout the text

Line 195: Same as line 194 - corrected

Reviewer 4 Report

Please find the following comments:

1. Novelty and unique contribution to knowledge that is transferrable beyond the specific case study: Please discuss how the findings of the work could apply beyond the region you considered, with a focus on developing countries, positioning this within the existing literature.

2. Pesticide use is rather broad and it remains unclear from reading: do the authors care about use at all, specific use? How to distinguish between reduced use by flower stripes as was asked in the survey to meet the aim of specific/sound/sustiainble use and to use at all? In this regard the authors need to be much more specific, what is their aim?

3. Methodology - do provide each model with how many farmers were interviewed?

4. The authors are advised to consider the respondents' characteristics as determinants of their Intension to follow flower strip cultivation to reduce pesticides usage in rice ecosystem.

5. The questionnaire used should be given as supplementary table?

6. Explain about Cronbach's alpha was tested for each construct in the questionnaire?

7. Discussion was too lengthy and it may be reduced and to the point that needs to be discussed. Address the above suggestion to improve the manuscript.

Author Response

We thank the reviewer for his/her thorough reading of the manuscripts and valuable suggestions to improve the text. We have responded to all the comments and made relevant changes as indicated here:

  1. Novelty and unique contribution to knowledge that is transferrable beyond the specific case study: Please discuss how the findings of the work could apply beyond the region you considered, with a focus on developing countries, positioning this within the existing literature. – we have now added text to the introduction (last paragraph) and discussion to indicate how this research can be applied beyond this case study. We also indicate the following existing text that also fulfills this purpose: [975-978]
  2. Pesticide use is rather broad and it remains unclear from reading: do the authors care about use at all, specific use? How to distinguish between reduced use by flower stripes as was asked in the survey to meet the aim of specific/sound/sustainable use and to use at all? In this regard the authors need to be much more specific, what is their aim? – we are clear in the text that, based on considerable field experience, we feel that chemical pesticides do not increase farmers yields and, indeed, only reduce profitability in tropical rice ecosystems. This position is polemic and will be criticized, therefore, to maintain some practical neutrality, we indicate that adherence to IPM principals would reduce insecticide use, or eliminate it altogether [see model development in section 2.1]. We have now, also, clarified this in the introduction [at the end of the first paragraph]. For further details on these aspects, see lines 161-163, 164, 165-167, 169-171, 628-630 [referring to reducing pesticides, but eliminating insecticides], 633-639, 770-774 [referring to other pesticides also], 868-870, 930-932.
  3. Methodology - do provide each model with how many farmers were interviewed? - in response to the reviewer’s comment, we have included as part of Table S1 the numbers of valid responses by farmers. Also please note that we have indicated the DFs for each model as footnotes to each of the analyses for Tables 1-3.
  4. The authors are advised to consider the respondents' characteristics as determinants of their Intension to follow flower strip cultivation to reduce pesticides usage in rice ecosystem. – this is the second paper from the same questionnaires – the first paper (Horgan et al 2022) addressed in considerable detail the reasons for which farmers adopted the intervention; and includes information on why some farmers actually abandoned ecological engineering. We have now added text to draw readers to that paper, and include some of the main results in the main text as regards farmers decisions to adopt or not  [see lines 328, 414-416]. Furthermore, in the context of the current paper, we included Table S2, which compared aspects of the farmers’ profiles for ecological engineering and conventional farmers. This information has been summarized in section 3.1..
  5. The questionnaire used should be given as supplementary table? – we have now included an English translation of the main questions or data collected with the supplementary information (Table S1) and included the citation in line 338. The table also includes the numbers of valid answers associated with each question to further respond to the reviewer’s’ comment no. 3.
  6. Explain about Cronbach's alpha was tested for each construct in the questionnaire? – We applied correspondence tests during the pre-testing stage mainly relating to questions that have been handled in the first paper from the survey (which included categorized responses to standard questions). The present paper mainly concerns continuous variables that are assessed for homogeneity after testing; and results from PERMANOVA analyses that test for internal relatedness during the ordination process and using real data from the final questionnaire (as opposed to the pre-testing responses). Because of the specific types of analyses conducted for the present paper, we have not included reference to the handling of pre-test data because the specific tests were not applied to the majority of the questions addressed here; however, we are grateful to the reviewer for reminding us to include the test-statistic thresholds in future papers with large numbers of categorical responses. See also 334-335 for further explanation.
  7. Discussion was too lengthy and it may be reduced and to the point that needs to be discussed. – we have reduced the length of the discussion where possible and following the reviewer’s recommendation; the current discussion is about 75% of the length of the original.